# BAYESIAN DECISION TREES FOR CONFOUNDER SELECTION IN MEDIATION ANALYSIS

## ABSTRACT

The estimation of causal effects in observational research fundamentally relies on proper adjustment for confounding variables. As such, identifying relevant confounders from the data is an important preliminary task. Although numerous data-driven techniques have been suggested for confounder selection in conventional exposure-outcome analyses, such methodologies are absent in causal mediation analysis, which entails identifying the effects within the exposure-mediator-outcome framework. This paper presents a Bayesian framework for confounder selection in mediation analysis via Bayesian additive regression trees (BART). We specify separate models for the exposure, mediator, and outcome, and introduce a common sparsity-inducing prior on their selection probability vectors. This enables identification of covariates that are jointly important across all models (that is, potential confounders). Furthermore, we introduce a novel criterion for confounder selection in the context of mediation analysis and establish that satisfaction of this criterion ensures the validity of the sequential ignorability assumption under certain conditions. The proposed method demonstrates consistently strong performance across a range of simulation scenarios, offering a practical approach for confounder selection in high-dimensional mediation analysis.

## 1 INTRODUCTION

Causal inference is a critical framework for identifying and understanding the underlying causes of observable events or outcomes, and it is widely applied in fields such as social science and health research. In observational studies, accurate estimation of causal effects requires careful adjustment for confounding variables that simultaneously influence both the exposure and the outcome. To address this challenge, various methods and assumptions have been proposed, including matching methods (Rosenbaum, 1989; Stuart, 2010; Rosenbaum, 2020), structural models (Sobel, 1996; Robins et al., 2000; Pearl, 2012), nonparametric approaches (Benkeser et al., 2017; Kennedy, 2019), and Bayesian inference frameworks (Hill, 2011; Li et al., 2023), among others.

The number of potential confounding variables to consider increases with the size of the data. In such cases, identifying an appropriate subset of confounders is an important first step before applying causal inference methods. A great deal of research has been conducted on confounder selection methodologies. Loh & Vansteelandt (2021) proposed a method to identify an optimal set of confounders based on covariates included in propensity score models, with the aim of achieving stable causal effect estimates. Hagstrom (2018) proposed using Markov and Bayesian networks to select confounders. Shortreed & Ertefaie (2017) introduced the outcome-adaptive LASSO technique to select appropriate confounders for inclusion in propensity score models. Shi et al. (2019) used neural networks to identify confounders relevant to both propensity score and outcome models. Wang et al. (2012), Lefebvre et al. (2014), and Wang et al. (2015) developed Bayesian inference methods for selecting variables for propensity score and outcome models.

When a mediator lies on the causal pathway between an exposure and an outcome, mediation analysis decomposes the total effect into direct and indirect effects, offering insight into causal mechanisms. Unlike standard causal analyses that adjust for confounders between treatment and outcome, mediation analysis must address three sets of confounders: those affecting the treatment-mediator $(A - M)$, treatment-outcome $(A - Y)$, and mediator-outcome $(M - Y)$ relationships, as shown in Figure 1. To address these challenges, several assumptions and methodological frameworks have

been proposed; detailed discussions can be found in Robins & Greenland (1992); Imai et al. (2010b); Pearl (2022).

In mediation analysis, selecting an appropriate set of confounders from a high-dimensional pool of candidate covariates is an crucial preliminary step. This task is particularly important and must be conducted with great care, as confounding bias can arise simultaneously across three distinct relationships. In general, confounders are selected based on domain knowledge or prior studies. However, to our knowledge, data-driven confounder selection methods specifically tailored for mediation analysis have not yet been thoroughly explored.

In this research, we developed new criteria for confounder selection specifically designed for mediation analysis. These criteria guide the identification of confounder sets that align with standard identification assumptions, such as sequential ignorability, especially in the presence of high-dimensional covariates. Building on these criteria, we propose a data-driven confounder selection method based on a Bayesian

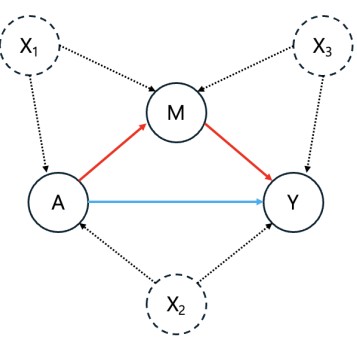

Figure 1: Directed acyclic graph (DAG) for mediation analysis. The red solid path (through the mediator $M$) denotes the indirect effect and the blue solid path denotes the direct effect. Dashed lines explain confounding relationships among $A$, $M$ and $Y$.

decision tree framework. Specifically, we use separate Bayesian additive regression trees (BART) priors on the treatment, mediator, and outcome models. We also impose a common prior on the variable selection probabilities across these models. This structure helps identify variables that are consistently important across all three models, which are the confounders.

1. We develop explicit confounder selection criteria grounded in the mediation analysis framework and aligned with key identification assumptions, especially when high-dimensional covariates are present.

2. We present a novel Bayesian methodology for data-driven confounder selection and show that it enhances the precision of causal effect estimation.

**Related work.** Vander Weele & Shpitser (2011) and VanderWeele (2019) proposed criteria for confounder selection in standard causal analyses focused on treatment–outcome relationships. However, in mediation analysis-where confounding must be addressed across three distinct causal pathways (as illustrated in Figure 1)-clear and explicit criteria for confounder selection have yet to be established. We seek to address this gap. Our work also extends the approach introduced by Kim et al. (2023), who applied Bayesian additive regression trees (BART; Chipman et al. (2010)) with a common prior on selection probabilities to identify confounders in treatment-outcome analyses. In contrast, we develop a method based on Bayesian causal mediation forests, a variant of Bayesian causal forests (Hahn et al., 2020), to jointly model the treatment, mediator, and outcome. By imposing a common prior across these three models, our method promotes confounder identification and helps more accurate estimation of causal effects.

## 2 NOTATION AND ESTIMANDS

Beyond estimating causal effect sizes, causal mediation analysis provides insight into the underlying mechanisms by decomposing the total effect of an exposure ($A$) on an outcome ($Y$) into distinct components (Imai et al., 2010a; VanderWeele, 2015). As shown in Figure 1, the total effect is partitioned into an *indirect effect* (red), which operates through a mediator ($M$), and a *direct effect* (blue), which captures the influence of $A$ on $Y$ not mediated by $M$. This decomposition enables attribution of the outcome variation to specific causal pathways, offering a more granular understanding of the relationship between $A$ and $Y$. Let $A_i$, $M_i$, and $Y_i$ denote the observed exposure, mediator, and outcome, respectively, for individual $i$, and let $\boldsymbol{X}_i$ be $P$-dimensional vector of pre-treatment covariates. In the potential outcomes framework (Rubin, 1974), $M_i(a)$ denote the potential mediator value that

would be observed for subset $i$ under exposure level $a$, and $Y_i(a, m)$ denotes the potential outcome under exposure $a$ and mediator value $m$. Under the extended Stable Unit Treatment Value Assumption (SUTVA) for mediation (Forastiere et al., 2016) which assumes (1) no interference between units and (2) no multiple versions of treatment, the observed variables can be expressed in terms of potential outcomes as: $M_i = M_i(A_i), \quad Y_i = Y_i(A_i, M_i(A_i))$.

A key challenge in mediation analysis is identifying unobserved potential outcomes such as $M_i(1 - A_i)$, $Y_i(1 - A_i, M_i(1 - A_i))$, and the cross-world quantity $Y_i(a, M_i(a'))$ for $a \neq a'$. Imai et al. (2010b) address this using the sequential ignorability assumption:

$$\{Y_i(a', m), M_i(a)\} \perp A_i \mid \boldsymbol{X}_i = \boldsymbol{x}, \tag{1}$$

$$Y_i(a', m) \perp M_i(a) \mid A_i = a, \boldsymbol{X}_i = \boldsymbol{x}, \tag{2}$$

for all $a, a' \in \{0, 1\}$, along with positivity $0 < \Pr(A_i = a \mid \boldsymbol{X}_i = \boldsymbol{x}), 0 < p(M_i(a) = m \mid A_i = a, \boldsymbol{X}_i = \boldsymbol{x})$ for all $m \in \mathcal{M}$ and $x \in \mathcal{X}$.

In the DAG (Fig. 1), Assumption Eq. (1) requires adjustment for exposure-mediator and exposure-outcome confounders (e.g., $X_1, X_2$), while Assumption Eq. (2) pertains to mediator-outcome confounders unaffected by exposure (e.g., $X_3$). A unified criterion for selecting covariates satisfying both remains underexplored; we address this in the next section.

Under these assumptions, $Y_i(a, M_i(a'))$ is identifiable. For units with covariates $\boldsymbol{X}_i = \boldsymbol{x}$, we define:

$$\zeta(\boldsymbol{x}) = \mathbb{E}[Y_i(1, M_i(1)) \mid \boldsymbol{X}_i = \boldsymbol{x}] - \mathbb{E}[Y_i(1, M_i(0)) \mid \boldsymbol{X}_i = \boldsymbol{x}],$$

$$\delta(\boldsymbol{x}) = \mathbb{E}[Y_i(1, M_i(0)) \mid \boldsymbol{X}_i = \boldsymbol{x}] - \mathbb{E}[Y_i(0, M_i(0)) \mid \boldsymbol{X}_i = \boldsymbol{x}].$$

Here, $\zeta(\boldsymbol{x})$ denotes the *indirect effect*, representing the effect mediated through $M_i$, while $\delta(\boldsymbol{x})$ denotes the *direct effect*, not operating through the mediator. Their sum yields the *total effect*, and averaging over $\boldsymbol{X}_i$ gives the marginal direct and indirect effects.

## 3 MEDIATION SPECIFIC CONFOUNDER SELECTION CRITERION

Confounder selection is fundamental to causal inference. While domain-driven selection is common, it becomes impractical in high-dimensional settings, prompting the development of formal criteria for exposure-outcome analyses. Notable examples include: (a) the pretreatment criterion, which adjusts for all covariates measured prior to treatment; (b) the common cause criterion, which adjusts only for variables affecting both exposure and outcome; (c) the disjunctive cause criterion, which includes any variable that is a cause of the exposure, the outcome, or both; (d) in addition, the recommendation to exclude instrumental variables to avoid $Z$-bias amplificationVanderWeele (2019).

However, these criteria are not designed for causal structures involving a mediator. To our knowledge, no formal criterion has been proposed for selecting confounders in mediation analysis. We address this gap by introducing a novel criterion that explicitly incorporates the mediator and supports identification under the sequential ignorability assumption.

**Definition 1** (Mediation Disjunctive Cause Criterion). *A covariate set $\mathcal{C}$ satisfies the mediation disjunctive cause criterion if it includes all variables that are direct causes of at least one of the following: the exposure $A$, the mediator $M$, or the outcome $Y$, excluding any variable that functions solely as an instrument-i.e., causes $A$ but not $M$ or $Y$ given $A$, or causes $M$ but not $Y$ given $M$.*

This criterion generalizes the traditional disjunctive cause criterion by accounting for the mediator $M$. Based on this definition, we establish the following result:

**Theorem 1.** *Suppose the set of observed pre-treatment covariates includes at least one direct cause for each backdoor path among $A$, $M$, and $Y$. Then, there exists some subsets $\mathcal{C}$ that satisfy the mediation disjunctive cause criterion and are sufficient for the sequential ignorability assumptions in Eqs. (1) and (2).*

The proof is provided in the Appendix. In essence, if the high-dimensional data include sufficient covariates to $d$-separate all backdoor paths among exposure, mediator, and outcome (Pearl, 2009), then some valid adjustment sets $\mathcal{C}$ can be constructed to satisfy sequential ignorability. In the next section, we propose a data-driven method to identify one of the possible sets $\mathcal{C}$ that satisfy the mediation disjunctive cause criterion by favoring a sparse set of covariates.

## 4 BART FOR CAUSAL MEDIATION ANALYSIS

To identify confounders in high-dimensional mediation settings, we adopt a Bayesian nonparametric approach based on Bayesian Additive Regression Trees (BART) (Chipman et al., 2010). BART models the outcome as a sum over $T$ regression trees:

$$Y_i = f(\boldsymbol{X}_i) + \epsilon_i, \quad \epsilon_i \sim \mathcal{N}(0, \sigma^2), \qquad f(\boldsymbol{X}_i) = \sum_{t=1}^{T} g(\boldsymbol{X}_i; \mathcal{T}_t, \mathcal{M}_t),$$

where each $g(\cdot)$ is a regression tree with structure $\mathcal{T}_t$ and terminal node parameters $\mathcal{M}_t$. Each tree comprises internal nodes consisting of splitting rules of the form $X_j < c$, where $X_j$ is a selected splitting variable from the pool. Trees are typically shallow and act as weak learners. BART updates tree structures via Bayesian backfitting (Hastie & Tibshirani, 2000), iteratively modifying each tree based on residuals from the others. Tree updates randomly apply one of four alterations: grow, prune, change, or swap. In grow and change steps, splitting variables are drawn according to a probability vector $\boldsymbol{s} = (s_1, \ldots, s_P)$ over the $P$ covariates. While a uniform prior is common for $\boldsymbol{s}$, sparsity-inducing priors improve variable selection (Linero, 2018). We adopt a symmetric Dirichlet prior: $\boldsymbol{s} \sim \text{Dir}\left(\frac{\alpha}{P}, \ldots, \frac{\alpha}{P}\right)$, which yields a simple conjugate update and encourages concentrated variable selection, promoting the identification of key confounders.

### 4.1 OBSERVED DATA MODEL FOR MEDIATION

Extending the BART framework, we model the exposure, mediator, and outcome as follows:

$$\Pr(A_i = 1) = \Phi\left\{\mu(\boldsymbol{X}_i)\right\}, \tag{3}$$

$$M_i = \mu_M(\boldsymbol{X}_i) + \tau_M(\boldsymbol{X}_i)A_i + \epsilon_i, \quad \epsilon_i \sim \mathcal{N}(0, \sigma_M^2), \tag{4}$$

$$Y_i = \mu_Y(\boldsymbol{X}_i) + \tau_Y(\boldsymbol{X}_i)A_i + \tau_Y'(\boldsymbol{X}_i)M_i + \eta_i, \quad \eta_i \sim \mathcal{N}(0, \sigma_Y^2), \tag{5}$$

where $\Phi(\cdot)$ denotes the standard normal CDF. Independent BART priors are placed on all functions: $\mu, \mu_M, \tau_M, \mu_Y, \tau_Y, \tau_Y'$.

The mediator model Eq. (4) follows the Bayesian Causal Forest (BCF) of Hahn et al. (2020), where $\tau_M(\boldsymbol{x})$ captures the conditional effect of exposure on the mediator, and $\mu_M(\boldsymbol{x})$ adjusts for confounding. The outcome model Eq. (5) builds on the Bayesian Causal Mediation Forest (BCMF) of Ting & Linero (2025), with $\tau_Y(\boldsymbol{x})$ representing the conditional direct effect $\zeta(\boldsymbol{x})$, and the product $\tau_Y'(\boldsymbol{x}) \cdot \tau_M(\boldsymbol{x})$ corresponding to the conditional indirect effect $\delta(\boldsymbol{x})$. The functions $\mu_Y(\boldsymbol{x})$ and $\mu_M(\boldsymbol{x})$ serve as prognostic terms to reduce confounding in the outcome model. For prognostic functions, we use a relatively large number of trees (e.g., 200) to capture complex covariate relationships and mitigate bias. For modifier functions ($\tau_M, \tau_Y, \tau_Y'$), we use fewer trees (e.g., 20) with priors that shrink toward homogeneity, discouraging unnecessary variable inclusion.

**Regularization Induced Confounding** Hahn et al. (2020) proposed augmenting the BART prior for the prognostic function in BCF with estimated propensity scores to mitigate *regularization-induced confounding*, i.e., bias introduced by regularization in predictive models. In the context of mediation, Ting & Linero (2025) similarly addressed this issue, also referred to as *prior dogmatism* (Linero, 2024), by introducing clever covariates $\hat{m}_{ai}$ for $a \in \{0, 1\}$ into the outcome model. The modified mediator and outcome models incorporating these bias-reducing covariates are:

$$M_i = \mu_M(\boldsymbol{X}_i, \hat{\pi}_i) + \tau_M(\boldsymbol{X}_i)A_i + \epsilon_i, \tag{6}$$

$$Y_i = \mu_Y(\boldsymbol{X}_i, \hat{\pi}_i, \hat{m}_{0i}, \hat{m}_{1i}) + \tau_Y(\boldsymbol{X}_i)A_i + \tau_Y'(\boldsymbol{X}_i)M_i + \eta_i, \tag{7}$$

where $\hat{\pi}_i = \hat{\mathbb{E}}(A_i \mid \boldsymbol{X}_i)$ is the estimated propensity score, and $\hat{m}_{ai} = \hat{\mathbb{E}}(M_i \mid A_i = a, \boldsymbol{X}_i)$ are the estimated conditional expectations of the mediator under each treatment level.

### 4.2 JOINT CONFOUNDER SELECTION VIA REPARAMETERIZED BART PRIORS

To perform confounder selection across the exposure, mediator, and outcome models in Eqs.(3), (6), and (7), we propose a unified approach linking the prognostic functions $\mu$, $\mu_M$, and $\mu_Y$. Each function is modeled with a BART prior, where internal nodes split on covariates drawn according

to a selection probability vector $\boldsymbol{s}$. By assigning a common sparsity-inducing Dirichlet prior to the selection probabilities, we jointly identify covariates that are consistently important for confounding adjustment-i.e., true confounders.

In models Eqs. (6) and (7), additional variables $(\hat{\pi}_i, \hat{m}_{0i}, \hat{m}_{1i})$ are included as inputs, each with associated selection probabilities: $s_\pi$ for the mediator model, and $s_\pi, s_{m0}, s_{m1}$ for the outcome model. As such, the three models involve distinct but overlapping selection vectors: $\boldsymbol{s}^A, \boldsymbol{s}^M, \boldsymbol{s}^Y$, making a naïve prior specification nontrivial.

We resolve this by reparameterizing all three vectors in terms of a common selection vector $\boldsymbol{s}^Y = \boldsymbol{s} = (s_1, \ldots, s_P, s_\pi, s_{m0}, s_{m1})$, and defining:

$$\boldsymbol{s}^A = \left( \frac{s_1}{1 - (s_\pi + s_{m0} + s_{m1})}, \ldots, \frac{s_P}{1 - (s_\pi + s_{m0} + s_{m1})} \right),$$

$$\boldsymbol{s}^M = \left( \frac{s_1}{1 - (s_{m0} + s_{m1})}, \ldots, \frac{s_P}{1 - (s_{m0} + s_{m1})}, \frac{s_\pi}{1 - (s_{m0} + s_{m1})} \right),$$

$$\boldsymbol{s}^Y = (s_1, \ldots, s_P, s_\pi, s_{m0}, s_{m1}).$$

This reparameterization leverages the neutrality property of the Dirichlet distribution. We impose the prior $\boldsymbol{s} \sim \mathrm{Dir}(\alpha/(P+3), \ldots, \alpha/(P+3))$, which induces consistent sparsity across all models.

Given this formulation, the posterior update of $\boldsymbol{s}$ is proportional to:

$$\mathcal{Q} = \prod_{t=1}^{T} \left\{ \prod_{b \in \mathcal{T}_t^\mu} \frac{s_{j_b}}{1 - (s_\pi + s_{m0} + s_{m1})} \cdot \prod_{c \in \mathcal{T}_t^{\mu_M}} \frac{s_{j_c}}{1 - (s_{m0} + s_{m1})} \cdot \prod_{e \in \mathcal{T}_t^{\mu_Y}} s_{j_e} \right\}$$

$$\times \left\{ (s_\pi s_{m0} s_{m1})^{\alpha/(P+3)-1} \cdot \prod_{j=1}^{P} s_j^{\alpha/(P+3)-1} \right\}, \tag{8}$$

where $\mathcal{T}_t^\mu$, $\mathcal{T}_t^{\mu_M}$, and $\mathcal{T}_t^{\mu_Y}$ denote the sets of internal nodes used in trees associated with $\mu$, $\mu_M$, and $\mu_Y$, respectively. In the Appendix, Algorithm 1 outlines the posterior sampling procedure. This sparsity-inducing prior encourages the selection of a minimal subset among the multiple possible covariate sets $\mathcal{C}$ that satisfy the mediation disjunctive cause criterion. Specifically, it tends to identify the variables with the strongest signals along backdoor paths, and as a result, selects direct causes rather than upstream ancestors. We present an empirical experiment demonstrating this behavior in Appendix F (Table A2 and Figure A4).

## 5 THEORETICAL PROPERTIES

Let $\mathcal{D} = \{(A_i, M_i, Y_i, \boldsymbol{X}_i)\}_{i=1}^{N}$ denote the observed dataset, where $A_i$ is the exposure, $M_i$ the mediator, $Y_i$ the outcome, and $\boldsymbol{X}_i = (X_{i1}, \ldots, X_{iP})$ is a $P$-dimensional vector of covariates.

**Theorem 2** (Consistency of Confounder Selection). *Assume models for exposure $A$, mediator $M$, and outcome $Y$ are correctly specified, and let $\mathcal{C}$ be one of the covariate sets satisfying the mediation disjunctive cause criterion. Suppose $P$ is fixed and the Dirichlet concentration parameter satisfies $\alpha = O(N^{-\gamma})$ for some $\gamma > 0$, with $\alpha/P \ll 1$. Then, as $N \to \infty$, for any $\epsilon \downarrow 0$,*

$$P(s_j > \epsilon \mid \mathcal{D}) \to \begin{cases} 1, & X_j \in \mathcal{C}, \\ 0, & X_j \notin \mathcal{C}, \end{cases}$$

*where $s_j$ denotes the posterior selection probability for $X_j$, and $\mathcal{D}$ denotes the observed data.*

The consistency is also guaranteed under the asymptotic growth condition $P = O(N^\gamma)$ with a properly controlled Dirichlet concentration parameter $\alpha$. The proof is provided in Appendix B.

**Corollary 1** (Consistency of Confounder Selection in High-Dimensional Settings $(P > N)$). *Assume models for exposure $A$, mediator $M$, and outcome $Y$ are correctly specified, and let $\mathcal{C}$ be one of the covariate sets satisfying the mediation disjunctive cause criterion. Further, suppose the number of covariates grows at the rate $P = O(N^\gamma)$ for some $\gamma > 1$ while the Dirichlet concentration parameter either grows more slowly than $P$ or decreases. Then, for any $\epsilon \downarrow 0$, as $N \to \infty$*

$$P(s_j > \epsilon \mid \mathcal{D}) \to \begin{cases} 1, & X_j \in \mathcal{C}, \\ 0, & X_j \notin \mathcal{C}, \end{cases}$$

*Proof.* This proof requires only one modification from the proof of Theorem 2. Specifically, when $\alpha$ grows at most at the rate of $O(N^\beta)$, it can be shown that $P(s_j > \epsilon | \mathcal{D})$ shrinks polynomially with $\alpha/(P+3) = O(N^\beta/N^\gamma)$ where $\beta < \gamma$. The cases where $\alpha$ is fixed or decays are straightforward. $\square$

**Definition 2.** *Let $\delta(x)$ and $\zeta(x)$ denote the true conditional indirect and direct effects given $X = x$, respectively. Let $\hat{\delta}(x)$ and $\hat{\zeta}(x)$ represent their corresponding posterior expectations under the observed data model. The total error is defined as*

$$\Delta_{total} = \sup_{x \in \mathcal{X}} |\hat{\delta}(x) - \delta(x)| + \sup_{x \in \mathcal{X}} |\hat{\zeta}(x) - \zeta(x)|.$$

**Definition 3.** *Let $\mathcal{S}$ represent any subset of $\{X_1, X_2, \ldots, X_P\}$. Define a neighborhood of a certain set $\mathcal{C}$ as*
$$\mathcal{N}_\epsilon(\mathcal{C}) = \{\mathcal{S} : d(\mathcal{S}, \mathcal{C}) < \epsilon\},$$
*where $d(\mathcal{S}, \mathcal{C})$ is a distance metric, measuring the number of incorrect inclusions or exclusions in $\mathcal{S}$ relative to $\mathcal{C}$ such as $d(\mathcal{S}, \mathcal{C}) = |\mathcal{S} \setminus \mathcal{C}| + |\mathcal{C} \setminus \mathcal{S}|$, and $\epsilon \geq 0$ controls the allowable "error" in selecting confounders.*

**Theorem 3** (Posterior Consistency of Mediation Effects). *Suppose the following conditions hold:*

1. *The models for the mediator and outcome are correctly specified.*

2. *The posterior distribution over the selected adjustment set $\mathcal{S}$ concentrates around a set $\mathcal{C}$ that satisfies the mediation disjunctive cause criterion, i.e., for any fixed $\epsilon > 0$,*

$$P(\mathcal{S} \in \mathcal{N}_\epsilon(\mathcal{C}) \mid \mathcal{D}) \to 1 \quad as \ N \to \infty.$$

3. *The mappings from the regression functions $(f_M, f_Y)$ to $(\delta(x), \zeta(x))$ are Lipschitz continuous with respect to the sup-norm. That is, there exists a constant $L > 0$ such that:*

$$|\delta_1(x) - \delta_2(x)| + |\zeta_1(x) - \zeta_2(x)| \leq L \cdot (\|f_{M,1} - f_{M,2}\|_\infty + \|f_{Y,1} - f_{Y,2}\|_\infty),$$

*for all $x \in \mathcal{X}$, and any pair $(f_{M,1}, f_{Y,1})$, $(f_{M,2}, f_{Y,2})$ obtained from BART posteriors for the mediator and outcome models, respectively, under confounder adjustment sets $\mathcal{S}_1$, $\mathcal{S}_2$.*

*Then, for any $\epsilon > 0$,*
$$P(\Delta_{total} > \epsilon \mid \mathcal{D}) \to 0 \quad as \ N \to \infty.$$

The proof is provided in the appendix.

*Remark*: Theorem 2 and Corollary 1 show that the posterior distribution of the selection probabilities, which drive confounder inclusion during the model updates for the exposure, mediator, and outcome, concentrates around sets that satisfy the mediation disjunctive cause criterion. This ensures that, asymptotically, the selected variables capture all necessary pathways for causal adjustment. Building on this, Theorem 3 demonstrates that, under mild regularity conditions, the posterior estimates of the indirect and direct effects converge to the corresponding true effects. These results collectively establish a principled link between posterior confounder selection and valid causal inference in mediation analysis.

## 6 Experiments

We evaluated the performance of the proposed model across a range of simulation scenarios with $P = 100$ covariates $\boldsymbol{X}$ and a sample size of $N = 250$. Among these, the first five covariates $X_1 - X_5$ were designated as true confounders. The simulation settings were designed to assess performance under diverse conditions, including heterogeneous treatment effects, non-linear mediator and outcome relationships, the presence of instrumental variables, and the inclusion of non-confounding predictors. The scenarios are summarized as follows: **Scenario 1**: Heterogeneous effects; non-linear mediator and outcome models, **Scenario 2**: Heterogeneous effects; non-linear models with two additional predictors in the outcome model, **Scenario 3**: Heterogeneous effects; non-linear models with two instrumental variables in the exposure model, **Scenario 4**: Heterogeneous effects;

non-linear models with two instrumental variables in the mediator model, **Scenarios 5 & 6**: High-dimensional settings ($P = 500 > N$) and ($P = 2000 > N$); otherwise identical to Scenario 2, **Scenario 7**: Low-dimensional setting ($P = 20$); otherwise identical to Scenario 2. Detailed specifications of the data-generating mechanisms for each scenario are provided in the Appendix.

To evaluate the performance of our proposed model (Bayesian causal mediation forest with confounder selection; **BCMF-CS**), we compare it against three baselines: (1) **BCMF**: A variant of our model based on Eqs. (6) and (7), using uniform (non-sparse) priors for each BART selection probability vector. (2) **DCMF**: A variant that assigns independent sparsity-inducing Dirichlet priors to each model component but does not enforce joint confounder selection. (3) **LSEM**: A classical linear structural equation model (Baron & Kenny, 1986; MacKinnon & Dwyer, 1993) fit with 5 true confounders and 5 additional covariates, to provide a slight advantage to LSEM, which lacks variable selection capability. BCMF follows the framework of Ting & Linero (2025) for heterogeneous mediation but assumes the confounder set is known a priori. DCMF extends this by allowing independent variable selection across models, while BCMF-CS introduces joint selection via a common prior. LSEM provides a parametric baseline for comparison. All methods are evaluated over $S = 200$ simulation replicates. Full specifications are provided in the Appendix.

We evaluate performance along two dimensions: effect estimation and confounder selection. For effect estimation, we report bias, absolute bias, mean squared error (MSE), and coverage, computed under heterogeneous effect settings. Each metric is computed per unit within a replicate and then averaged over $S = 200$ replicates. For example, the bias for the indirect effect is given by Bias $= \frac{1}{S} \sum_{s=1}^{S} \left\{ \frac{1}{n} \sum_{i=1}^{n} \left( \zeta(\boldsymbol{X}_i) - \hat{\zeta}(\boldsymbol{X}_i) \right) \right\}$. Confounder selection is evaluated using true positive rate (TPR), false positive rate (FPR), positive predictive value (PPV), and F1 score. A variable is considered selected if its posterior inclusion probability (PIP) exceeds 0.5(Barbieri & Berger, 2004), and comparisons are made against the ground truth separately for the exposure, mediator, and outcome models.

Table 1 reports the results for effect estimation. Across all scenarios, BCMF-CS consistently achieves superior performance in estimating causal effects. While DCMF shows similar performance in some scenarios, its accuracy notably deteriorates in Scenarios 2 and 3 when non-confounding predictors or instruments are introduced. In these settings, BCMF-CS outperforms DCMF in terms of absolute bias, MSE, and coverage. These improvements stem from BCMF-CS's ability to accurately identify true confounders.

Table A4 (Appendix H) summarizes confounder selection performance. Competing models such as BCMF and DCMF do not effectively select variables for the exposure model ($A$), resulting in lower performance in this component. Even in the mediator ($M$) and outcome ($Y$) models (more directly tied to effect estimation) BCMF-CS consistently outperforms all baselines across evaluation metrics.

Figures A2 and A3 (Appendix H) display posterior inclusion probabilities under Scenarios 2 and 4, which involve additional outcome predictors and mediator instruments, respectively. BCMF-CS reliably selects true confounders with high probability while suppressing irrelevant variables. Notably, in Figure A3, the exclusion of instrumental variables $X_8$ and $X_9$ from the outcome model illustrates BCMF-CS's ability to respect the mediation disjunctive cause criterion.

**Sensitivity Analysis and Additional Experiments**   To assess robustness to prior settings (particularly the number of trees) we conducted a sensitivity analysis based on Scenario 1. The main experiments (Table 1) used 200 trees for prognostic functions ($\mu, \mu_M, \mu_Y$) and 20 for modifier functions ($\tau_Y, \tau_Y'$); for comparison, we also tested 100 and 50 trees. Results in Table A1 show stable performance once a sufficient number of trees is used, consistent with Chipman et al. (2010). We further conducted additional analyses (Table A2 and Figure A4): (1) sensitivity to the Dirichlet concentration parameter, (2) the impact of unmeasured confounding, (3) the effect of model misspecification, and (4) identification of direct causes under multiple ancestors in the backdoor path.

**Computational Efficiency.**   Figure A1 summarizes the runtime of BCMF-CS based on 10,000 MCMC iterations executed on a Mac Studio (Apple M1, 128 GB RAM). We varied the number of observations $N$ and covariates $P$, using the tree configuration described in the Appendix. The runtime scales moderately with $N$, while remaining largely unaffected by $P$, reflecting the sparsity-

Table 1: Results (bias, absolute bias, mean squared error, and coverage) for different scenarios: Scenario 1 (base), Scenario 2 (with two predictors), Scenarios 3 and 4 (with two instruments in exposure and mediator models, respectively), Scenario 5 ($N < P = 500$), Scenario 6 ($N < P = 2000$), and Scenario 7 ($P = 20$). Results are reported separately for the direct effect (DE) and indirect effect (IE).

| Scenario | Method | BIAS | | Abs.BIAS | | MSE | | Coverage | |
|---|---|---|---|---|---|---|---|---|---|
| | | DE | IE | DE | IE | DE | IE | DE | IE |
| (1) | BCMF-CS | 0.048 | -0.024 | 0.172 | 0.248 | 0.056 | 0.167 | 0.99 | 0.90 |
| | BCMF | 0.128 | 0.016 | 0.308 | 0.447 | 0.179 | 0.415 | 0.92 | 0.69 |
| | DCMF | 0.023 | -0.001 | 0.170 | 0.338 | 0.052 | 0.259 | 0.98 | 0.76 |
| | LSEM | 1.091 | 0.213 | 1.100 | 1.075 | 1.438 | 1.717 | 0.19 | 0.39 |
| (2) | BCMF-CS | -0.029 | 0.039 | 0.431 | 0.469 | 0.303 | 0.429 | 0.77 | 0.64 |
| | BCMF | 0.069 | 0.133 | 0.712 | 0.856 | 0.737 | 1.159 | 0.62 | 0.17 |
| | DCMF | -0.073 | 0.120 | 0.782 | 0.814 | 0.936 | 1.108 | 0.53 | 0.22 |
| | LSEM | 1.038 | 0.212 | 1.075 | 1.072 | 1.629 | 1.709 | 0.52 | 0.38 |
| (3) | BCMF-CS | -0.038 | 0.038 | 0.308 | 0.488 | 0.167 | 0.454 | 0.77 | 0.59 |
| | BCMF | -0.028 | 0.123 | 0.599 | 0.839 | 0.464 | 1.123 | 0.52 | 0.17 |
| | DCMF | -0.098 | 0.119 | 0.704 | 0.811 | 0.785 | 1.096 | 0.47 | 0.22 |
| | LSEM | 0.928 | 0.158 | 0.947 | 1.047 | 1.121 | 1.649 | 0.54 | 0.39 |
| (4) | BCMF-CS | -0.014 | 0.036 | 0.144 | 0.390 | 0.050 | 0.323 | 0.94 | 0.72 |
| | BCMF | 0.019 | 0.110 | 0.464 | 0.763 | 0.307 | 0.969 | 0.70 | 0.31 |
| | DCMF | -0.011 | 0.031 | 0.211 | 0.434 | 0.133 | 0.411 | 0.87 | 0.67 |
| | LSEM | 0.945 | 0.196 | 0.960 | 1.069 | 1.149 | 1.704 | 0.53 | 0.39 |
| (5) | BCMF-CS | -0.016 | -0.021 | 0.087 | 0.332 | 0.022 | 0.234 | 0.99 | 0.80 |
| | BCMF | 0.217 | 0.035 | 0.402 | 0.993 | 0.463 | 1.596 | 0.85 | 0.18 |
| | DCMF | -0.012 | 0.001 | 0.437 | 0.617 | 0.553 | 0.808 | 0.73 | 0.49 |
| | LSEM | 1.024 | 0.265 | 1.041 | 1.169 | 1.357 | 2.056 | 0.54 | 0.44 |
| (6) | BCMF-CS | -0.058 | 0.031 | 0.058 | 0.429 | 0.004 | 0.345 | 1.00 | 0.62 |
| | BCMF | 0.186 | 0.144 | 0.358 | 1.105 | 0.422 | 1.787 | 0.82 | 0.06 |
| | DCMF | -0.122 | 0.132 | 0.878 | 1.004 | 0.956 | 1.486 | 0.36 | 0.02 |
| | LSEM | 0.944 | 0.067 | 1.104 | 1.062 | 1.820 | 1.765 | 0.52 | 0.34 |
| (7) | BCMF-CS | 0.001 | -0.008 | 0.267 | 0.444 | 0.149 | 0.379 | 0.87 | 0.69 |
| | BCMF | 0.215 | 0.038 | 0.402 | 0.988 | 0.451 | 1.583 | 0.86 | 0.19 |
| | DCMF | -0.017 | -0.007 | 0.397 | 0.583 | 0.485 | 0.749 | 0.76 | 0.53 |
| | LSEM | 1.011 | 0.272 | 1.030 | 1.169 | 1.342 | 2.051 | 0.55 | 0.43 |

inducing prior's effectiveness in focusing computation on relevant covariates. Overall, the method achieves practical scalability for moderate to large datasets.

## 7 REAL APPLICATION: AIDS CLINICAL TRIALS GROUP STUDY 175

The ACTG175 study (Hammer et al., 1996) is a randomized clinical trial evaluating antiretroviral therapies for HIV-positive patients, specifically comparing zidovudine alone, didanosine alone, and their combination. The dataset, available in the R package `speff2trial`, includes 2,139 patients and 27 variables. We define the outcome ($Y$) as CD4 T cell count at $95 \pm 5$ weeks and the mediator ($M$) as CD4 count at $20 \pm 5$ weeks. The binary exposure ($A$) distinguishes zidovudine only ($A = 0$) from the other regimens ($A = 1$). In addition, 15 pretreatment covariates (e.g., age, weight) are included, as detailed in the Appendix.

Figure 2 (left) displays estimated individual total, direct, and indirect effects. Direct effects exhibit limited heterogeneity, whereas indirect effects are clearly stratified by race (black circles: white; red triangles: non-white), suggesting that race contributes to heterogeneity in mediated pathways but not in direct effects. Figure 2 (right) shows posterior inclusion probabilities (PIPs) for confounders across the exposure, mediator, and outcome models. Two variables-baseline CD4 count (`cd40`) and prior antiretroviral duration (`preanti`)-are consistently selected (PIP = 1). Given the randomized design, exposure-mediator and exposure-outcome confounding is minimal, so these covariates most likely adjust for mediator–outcome confounding.

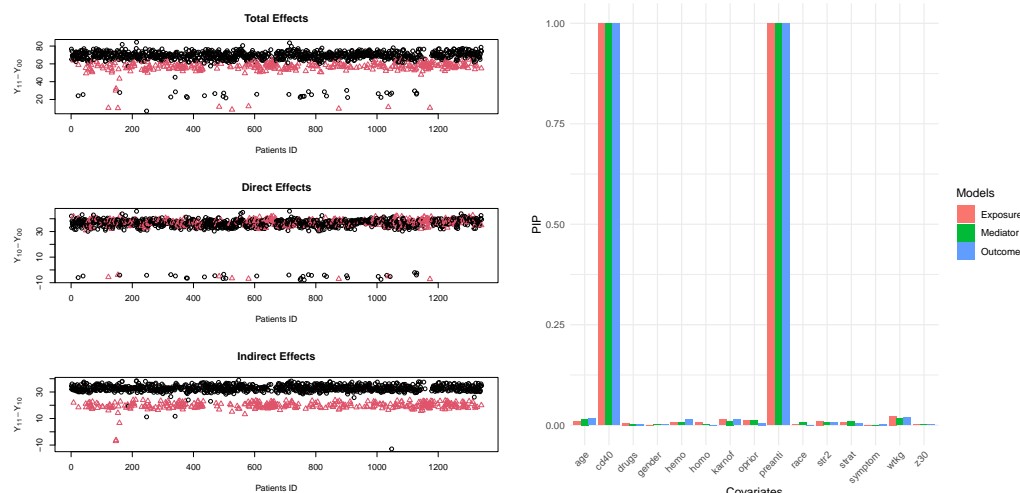

Figure 2: Results based on the ACTG175 dataset. (Left) Estimated individual total, direct, and indirect effects, ordered by patient ID. Red triangles represent non-white patients; black circles represent white patients. (Right) Posterior inclusion probabilities of potential confounders in the exposure, mediator, and outcome models.

It is important to note that although the mediator serves as a short-term proxy for the outcome, long-term effects (e.g., over a 75-week period) can still emerge, particularly depending on exposure status. Because the indirect effect reflects both the $A \to M$ and $M \to Y$ pathways, it may account for only part of the total effect when the $A \to M$ association is weak, even if the $M \to Y$ association is strong. Consequently, a direct effect may also be present. Similar delayed responses have been documented in prior studies (see *Guidelines for the Use of Antiretroviral Agents in Adults and Adolescents With HIV*, https://clinicalinfo.hiv.gov/).

## 8 LIMITATIONS AND FUTURE DIRECTIONS

In this study, we propose a Bayesian nonparametric approach for identifying confounders necessary for mediation analysis in potentially high-dimensional settings and for estimating individual (heterogeneous) direct and indirect effects. Specifically, we build on Bayesian additive regression trees by employing a variant known as the Bayesian causal mediation forest. By assigning a common sparsity-inducing prior to the selection probability vectors governing the prognostic functions of each model, proposed method identifies pretreatment covariates that are commonly used across models (i.e., the confounders). We establish theoretical properties for the proposed method, including posterior consistency of the confounder selection procedure, and demonstrate strong empirical performance across diverse simulation settings.

Formal criteria for confounder selection in mediation analysis are scarce. We introduce the mediation disjunctive cause criterion and show that it guarantees the sequential ignorability assumption, which is key to identifying mediation effects. This requires observing at least one direct cause for each backdoor path, an assumption that is strong but often plausible in high-dimensional settings with rich covariate information.

Another limitation of our study is its focus on the single-mediator setting. Extending the method to multiple or sequential mediators, which frequently arise in practice, is an important direction for future work. That said, the core algorithm extends naturally: in Appendix H we show how the model adapts to the case of multiple mediators (two mediators) and provide results from a toy example. Developing a fully general approach remains as future work.

## ETHICS STATEMENT (OPTIONAL)

This study does not involve human subjects and does not present any potentially harmful insights, methodologies, or applications. Furthermore, it does not raise issues related to potential conflicts of interest or sponsorship, discrimination, bias or fairness, privacy or security, legal compliance, or research integrity.

## REPRODUCIBILITY STATEMENT (OPTIONAL)

To ensure reproducibility of our study, we provide details in the Appendix that could not be fully included in the main manuscript. Specifically, we report the complete parameter and hyperparameter settings (Appendix C), present pseudocode to clearly illustrate how the method operates (Appendix D), and describe the data-generating processes for all examined experiments to facilitate replication of our results (Appendix E). In addition, we supply detailed proofs in Appendices A and B to establish the validity of the theorems stated in the main text.

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

# A   APPENDIX

## APPENDIX A: PROOF OF THEOREM 1

**Theorem 1.** *Suppose the set of observed pre-treatment covariates includes at least one direct cause for each backdoor path among $A$, $M$, and $Y$. Then, there exists some subsets $\mathcal{C}$ that satisfy the mediation disjunctive cause criterion and are sufficient for the sequential ignorability assumptions in Eqs. (1) and (2).*

*Proof.* Let $\mathcal{C}^\star$ denote a set of high-dimensional pre-treatment covariates that includes at least one direct cause for each backdoor path between the exposure $A$ and the mediator $M$, and also includes direct causes of $M$ that do not lie on $A - M$ backdoor paths. Let $C \subseteq C^\star$ be the subset consisting of variables that are causes of $A$, $M$, or both. Since high-dimensional pretreatment covariate set $C^\star$ contains at least one direct cause that lies on each backdoor path between $A$ and $M$, there exists a set $\mathcal{W} \subset \mathcal{C}^\star$ such that $\mathcal{W}$ blocks all backdoor paths from $A$ to $M$. Then, by Theorem 1 in Vander Weele & Shpitser (2011), $\mathcal{C}$ does also block all backdoor paths from $A$ to $M$ and thus $M(a) \perp A \mid \mathcal{C}$. Similarly, other conditional independence relationships can be shown based on the covariate set $\mathcal{C}$. $\qquad\square$

## APPENDIX B: PROOFS OF THEOREMS 2 AND 3

In Section 5, we stated several theoretical properties of the proposed method. This appendix provides detailed proofs of those results.

**Theorem 2** (Posterior Consistency of Confounder Selection)**.** *Assume models for exposure $A_i$, mediator $M_i$, and outcome $Y_i$ are correctly specified, and let $\mathcal{C}$ be one the covariate sets satisfying the mediation disjunctive cause criterion. Suppose $P$ is fixed and the Dirichlet concentration parameter satisfies $\alpha = O(N^{-\gamma})$ for some $\gamma > 0$, with $\alpha/P \ll 1$. Then, as $N \to \infty$, for any $\epsilon \downarrow 0$,*

$$P(s_j > \epsilon \mid \mathcal{D}) \to \begin{cases} 1, & X_j \in \mathcal{C}, \\ 0, & X_j \notin \mathcal{C}. \end{cases}$$

*Proof.* For the asymptotic consistency, we first examine the posterior probability of selecting a variable under the Dirichlet prior $\boldsymbol{s} \sim \text{Dir}\left(\frac{\alpha}{P+3}\right)$. The posterior selection probability is represented as

$$P(s_j > \epsilon \mid \mathcal{D}) = \frac{R_j P(s_j > \epsilon)}{R_j P(s_j > \epsilon) + P(s_j \leq \epsilon)},$$

where $R_j = \frac{L(\mathcal{D} \mid s_j > \epsilon)}{L(\mathcal{D} \mid s_j \leq \epsilon)}$ is the likelihood ratio comparing models with a higher versus lower probability of including the $j$-th covariate. The terms $P(s_j > \epsilon)$ and $P(s_j \leq \epsilon)$ depend on the Dirichlet prior specification.

For $X_j \in \mathcal{C}$ (i.e., a direct cause of $A$, $M$, or $Y$), exclusion of $X_j$ leads to a substantial increase in residual variance in at least one model. Under standard Gaussian likelihood assumptions, the log-likelihood ratio can be represented as

$$\log R_j = \log \frac{L(\mathcal{D} \mid X_j \text{ inclusion})}{L(\mathcal{D} \mid X_j \text{ exclusion})} = -\frac{N}{2} \log \frac{\sigma_{j,1}^2}{\sigma_{j,0}^2} - \frac{1}{2} \sum_{i=1}^{N} \left[ \frac{(y_i - \hat{y}_{i,1})^2}{\sigma_{j,1}^2} - \frac{(y_i - \hat{y}_{i,0})^2}{\sigma_{j,0}^2} \right].$$

where $\sigma_{j,0}^2$ and $\sigma_{j,1}^2$ are the total residual variances from the exposure, mediator and outcome models excluding and including $X_j$, respectively. By assuming $\frac{1}{N} \sum_i (y_i - \hat{y}_{i,1})^2 \to \sigma_{j,1}^2$ and $\frac{1}{N} \sum_i (y_i - \hat{y}_{i,0})^2 \to \sigma_{j,0}^2$ as $N \to \infty$,

$$\log R_j \to -\frac{N}{2} \log \frac{\sigma_{j,1}^2}{\sigma_{j,0}^2} - \frac{N}{2} \left( \frac{\sigma_{j,1}^2}{\sigma_{j,1}^2} - \frac{\sigma_{j,0}^2}{\sigma_{j,0}^2} \right) = -\frac{N}{2} \log \frac{\sigma_{j,1}^2}{\sigma_{j,0}^2}.$$

Since $\sigma_{j,1}^2 < \sigma_{j,0}^2$,

$$\log R_j = -\frac{N}{2} \log \frac{\sigma_{j,1}^2}{\sigma_{j,0}^2} = O(N) \quad \Rightarrow \quad R_j = \exp(O(N)).$$

For $X_j \notin \mathcal{C}$, trivially, $R_j \approx 1$ (no variance improvement).

Since the selection probabilities $\boldsymbol{s}$ are drawn from a Dirichlet prior, the marginal distribution of each component follows a Beta distribution $s_j \sim \text{Beta}\left(\frac{\alpha}{P+3}, \alpha - \frac{\alpha}{P+3}\right)$. The cumulative probability near zero for a single component is approximated by

$$P(s_j \leq \epsilon) = \frac{\Gamma(\alpha)}{\Gamma\left(\frac{\alpha}{P+3}\right)\Gamma\left(\alpha - \frac{\alpha}{P+3}\right)} \int_0^\epsilon s_j^{\frac{\alpha}{P+3}-1}(1-s_j)^{\alpha - \frac{\alpha}{P+3}-1} \, ds_j$$

$$= \frac{\Gamma(\alpha)}{\Gamma\left(\frac{\alpha}{P+3}\right)\Gamma\left(\alpha - \frac{\alpha}{P+3}\right)}(1 + o(1))\frac{\epsilon^{\frac{\alpha}{P+3}}}{\frac{\alpha}{P+3}} \qquad \left( \because (1-s_j)^{\alpha - \frac{\alpha}{P+3}-1} \to 1 \text{ as } \epsilon \to 0 \right).$$

Using the asymptotic behavior of the Gamma function near zero (the Laurent expansion of gamma function), $\Gamma(\alpha) \approx \frac{1}{\alpha}$ as $\alpha \to 0$ and $\Gamma\left(\alpha - \frac{\alpha}{P+3}\right) \approx \Gamma(\alpha)$ since $\alpha/(P+3)$ is small relative to $\alpha$, the probability of being non-trivially positive becomes:

$$P(s_j > \epsilon) \approx 1 - \frac{\Gamma(\alpha)}{\frac{P+3}{\alpha}\Gamma(\alpha)}(1 + o(1))\frac{\epsilon^{\frac{\alpha}{P+3}}}{\frac{\alpha}{P+3}}$$

$$= 1 - \epsilon^{\frac{\alpha}{P+3}}(1 + o(1)) = \frac{\alpha}{P+3}|\ln(\epsilon)| + O\left(\frac{\alpha^2}{(P+3)^2}\right).$$

For $X_j \in \mathcal{C}$, since $R_j$ grows exponentially as $N \to \infty$ while $P(s_j > \epsilon)$ shrinks polynomially with $\alpha = O(N^{-\gamma})$ for some $0 < \gamma < 1$ in

$$P(s_j > \epsilon \mid \mathcal{D}) = \frac{R_j P(s_j > \epsilon)}{R_j P(s_j > \epsilon) + P(s_j \leq \epsilon)},$$

the exponential growth in $R_j$ dominates over the shrinking prior as $N \to \infty$, yielding

$$P(s_j > \epsilon \mid \mathcal{D}) \to 1.$$

For $X_j \notin \mathcal{C}$, since $R_j \approx 1$

$$P(s_j > \epsilon \mid \mathcal{D}) \approx \frac{P(s_j > \epsilon)}{P(s_j > \epsilon) + P(s_j \leq \epsilon)}.$$

Since $P(s_j > \epsilon) = O\left(\frac{\alpha}{P+3}\right)$ and $P(s_j \leq \epsilon) = 1 - O\left(\frac{\alpha}{P+3}\right)$, we obtain

$$P(s_j > \epsilon \mid \mathcal{D}) = O\left(\frac{\alpha}{P+3}\right).$$

Thus, under the assumption $\alpha = O(N^{-\gamma})$,

$$P(s_j > \epsilon \,|\, \mathcal{D}) = O\left(\frac{1}{N^\gamma}\right) \to 0.$$

$\square$

**Theorem 3** (Posterior Consistency of Mediation Effects). *Suppose the following conditions hold:*

1. *The models for the mediator and outcome are correctly specified.*

2. *The posterior distribution over the selected adjustment set $\mathcal{S}$ concentrates around a set $\mathcal{C}$ that satisfies the mediation disjunctive cause criterion, i.e., for any fixed $\epsilon > 0$,*

$$P(\mathcal{S} \in \mathcal{N}_\epsilon(\mathcal{C}) \,|\, \mathcal{D}) \to 1 \quad as \ N \to \infty.$$

3. *The mappings from the regression functions $(f_M, f_Y)$ to $(\delta(x), \zeta(x))$ are Lipschitz continuous with respect to the sup-norm (uniform convergence). That is, there exists a constant $L > 0$ such that:*

$$|\delta_1(x) - \delta_2(x)| + |\zeta_1(x) - \zeta_2(x)| \leq L \cdot (\|f_{M,1} - f_{M,2}\|_\infty + \|f_{Y,1} - f_{Y,2}\|_\infty),$$

*for all $x \in \mathcal{X}$, and any pair $(f_{M,1}, f_{Y,1})$, $(f_{M,2}, f_{Y,2})$ obtained from BART posteriors for the mediator and outcome models, respectively, under confounder adjustment sets $\mathcal{S}_1$, $\mathcal{S}_2$.*

*Then, for any $\epsilon > 0$,*
$$P(\Delta_{total} > \epsilon \,|\, \mathcal{D}) \to 0 \quad as \ N \to \infty.$$

*Proof.* First, note that the posterior estimates of mediation effects at any $x$ are defined by integrating over the posterior distribution:s:

$$\hat{\delta}(x) = \int \delta(x; f_M, f_Y) \,\Pi(df_M, df_Y \,|\, \mathcal{D}), \quad \hat{\zeta}(x) = \int \zeta(x; f_M, f_Y) \,\Pi(df_M, df_Y \,|\, \mathcal{D}),$$

where the posterior distribution $\Pi(df_M, df_Y \,|\, \mathcal{D})$ is a mixture over adjustment sets $\mathcal{S}$:

$$\Pi(df_M, df_Y \,|\, \mathcal{D}) = \sum_{\mathcal{S} \subseteq \{1,\ldots,P\}} \Pi(df_M, df_Y \,|\, \mathcal{D}, \mathcal{S}) \cdot P(\mathcal{S} \,|\, \mathcal{D}).$$

Thus, we express the posterior error explicitly as:

$$\Delta_{\text{total}} = \sup_x \left| \sum_{\mathcal{S}} P(\mathcal{S} \,|\, \mathcal{D})(\hat{\delta}(x; \mathcal{S}) - \delta(x)) \right| + \sup_x \left| \sum_{\mathcal{S}} P(\mathcal{S} \,|\, \mathcal{D})(\hat{\zeta}(x; \mathcal{S}) - \zeta(x)) \right|$$

Then, we split the posterior sum over $\mathcal{S}$ into two disjoint parts:

(1) Adjustment sets $\mathcal{S} \in \mathcal{N}_\epsilon(\mathcal{C})$

By Assumption 2, such sets differ only slightly from the true confounder set $\mathcal{C}$. Because the models are correctly specified, the regression functions $(f_M, f_Y)$ trained on these nearly-correct $\mathcal{S}$ will be uniformly close to those trained on the true $\mathcal{C}$, in the sup-norm sense. Formally, for such $\mathcal{S}$,

$$\|f_M^{(\mathcal{S})} - f_M^{(\mathcal{C})}\|_\infty \to 0, \quad \|f_Y^{(\mathcal{S})} - f_Y^{(\mathcal{C})}\|_\infty \to 0 \quad as \ N \to \infty,$$

by posterior consistency of BART under correct model specification and near-correct covariate adjustment. Then, by Lipschitz continuity (Assumption 3), we obtain:

$$\sup_x |\delta^{(\mathcal{S})}(x) - \delta(x)| + \sup_x |\zeta^{(\mathcal{S})}(x) - \zeta(x)| \to 0.$$

Since the posterior concentrates on such $\mathcal{S}$ as $N \to \infty$, the contribution to $\Delta_{\text{posterior}}$ from this region vanishes.

(2) Adjustment sets $\mathcal{S} \notin \mathcal{N}_\epsilon(\mathcal{C})$

For these sets, by Assumption (2),

$$P(\mathcal{S} \notin \mathcal{N}_\epsilon(\mathcal{C}) \mid \mathcal{D}) \to 0.$$

Even if the pointwise error $\sup_x |\delta^{(\mathcal{S})}(x) - \delta(x)|$ or $\sup_x |\zeta^{(\mathcal{S})}(x) - \zeta(x)|$ is non-negligible (due to confounder omission or overfitting), the posterior weight on such $\mathcal{S}$ is exponentially small.

Therefore, the total posterior probability that such $\mathcal{S}$ dominate the posterior expectation vanishes:

$$\sum_{\mathcal{S} \notin \mathcal{N}_\epsilon(\mathcal{C})} P(\mathcal{S} \mid \mathcal{D}) \cdot \sup_x |\hat{\delta}(x; \mathcal{S}) - \delta(x)| \to 0.$$

Therefore, for any $\epsilon > 0$, we conclude:

$$P(\Delta_{\text{total}} > \epsilon \mid \mathcal{D}) \to 0 \quad \text{as } N \to \infty.$$

$\square$

APPENDIX C: PRIOR AND PARAMETER SETTING FOR THE BAYESIAN CAUSAL MEDIATION FOREST

The priors and parameter settings for the Bayesian causal mediation forest used in the main text are as follows:

- Number of trees ($T$): We used 200 trees for the prognostic functions and 20 trees for the modifier functions as default settings.
- Tree depth prior: We employed the prior $\alpha(1 + d)^{-\beta}$, where $\alpha \in (0, 1)$ and $\beta \in [0, \infty)$, for the depth $d$ of each tree. For prognostic functions, we set $\alpha = 0.95$ and $\beta = 2$; for modifier functions, we used $\alpha = 0.5$ and $\beta = 2$.
- Residual variance prior: The residual variance $\sigma^2$ for each model followed the prior $\sigma^2 \sim \nu\lambda/\chi_\nu^2$, with $\nu = 3$. The scale parameter $\lambda$ was chosen such that $P(\sigma < \hat{\sigma}) = 0.90$, where $\hat{\sigma}$ is a data-driven estimate.
- Tree alteration steps: While the original BART algorithm (Chipman et al., 2010) includes four tree alteration steps (grow, prune, change, and swap), we omit the swap step based on prior findings that sufficient mixing is achieved without it (Kapelner & Bleich, 2016). Hence, trees are updated using only the grow, prune, and change operations.

All other priors and parameter follow the default specifications provided in Chipman et al. (2010).

*BCMF model specification.* Under Models (3), (6), and (7), the BCMF model assigns independent discrete uniform priors over selection probabilities for each prognostic function:

$$\boldsymbol{s}^A = \left(s_1^A, s_2^A, \ldots, s_P^A\right) \sim \text{discrete uniform}(1, P),$$
$$\boldsymbol{s}^M = \left(s_1^M, s_2^M, \ldots, s_P^M, s_\pi^M\right) \sim \text{discrete uniform}(1, P+1),$$
$$\boldsymbol{s}^Y = \left(s_1^Y, s_2^Y, \ldots, s_P^Y, s_\pi^Y, s_{m0}, s_{m1}\right) \sim \text{discrete uniform}(1, P+3).$$

*DCMF model specification.* Under Models (3), (6), and (7), the DCMF model assigns independent sparsity-inducing Dirichlet priors over selection probabilities for each prognostic function:

$$\boldsymbol{s}^A = \left(s_1^A, s_2^A, \ldots, s_P^A\right) \sim \mathcal{D}\left(\frac{1}{P}, \cdots, \frac{1}{P}\right),$$
$$\boldsymbol{s}^M = \left(s_1^M, s_2^M, \ldots, s_P^M, s_\pi^M\right) \sim \mathcal{D}\left(\frac{1}{P+1}, \cdots, \frac{1}{P+1}\right),$$
$$\boldsymbol{s}^Y = \left(s_1^Y, s_2^Y, \ldots, s_P^Y, s_\pi^Y, s_{m0}, s_{m1}\right) \sim \mathcal{D}\left(\frac{1}{P+3}, \cdots, \frac{1}{P+3}\right).$$

*LSEM model specification.* This model adopts the classical linear structural equation framework:

$$Y_i(a, m) = \beta_{0Y} + \boldsymbol{X}_i^\top \beta_Y + \gamma_{0Y} a + \zeta_{0Y} m + \epsilon_i, \quad \epsilon_i \sim \mathcal{N}(0, \sigma_Y^2),$$
$$M_i(a) = \beta_{0M} + \boldsymbol{X}_i^\top \beta_M + \gamma_{0M} a + \nu_i, \quad \nu_i \sim \mathcal{N}(0, \sigma_M^2),$$

where $\boldsymbol{X}_i$ denotes the vector of covariates included for confounder adjustment.

APPENDIX D: PSEUDO-ALGORITHM

---

**Algorithm 1** Posterior Computation Algorithm

---

**Require:** Posterior samples from the previous iteration:

$$\phi^{\star} = \left[ \{ (\mathcal{T}_t^f, \mathcal{M}_t^f) \mid t = 1, \ldots, T_f; \ f \in \{\mu, \mu_M, \tau_M, \mu_Y, \tau_Y, \tau_Y'\} \}, \sigma_Y^2, \sigma_M^2, \boldsymbol{M}^{\text{mis}} \right]$$

1: **for** $r = 1, \ldots, R$ **do**                                    $\triangleright$ Repeat for $R$ MCMC iterations
2:    **for** $f \in \{\mu, \mu_M, \tau_M, \mu_Y, \tau_Y, \tau_Y'\}$ **do**
3:        **for** $t = 1, \ldots, T_f$ **do**
4:            **for** $i = 1, \ldots, n$ **do**
5:                Compute residual for observation $i$ in tree $t$ for function $f$:    $R_{i,-t}^{f,(r)}$
6:            **end for**
7:            Sample tree structure: $\mathcal{T}_t^{f,(r)} \sim p\left( \mathcal{T}_t^f \mid R_{1,-t}^{f,(r)}, \ldots, R_{n,-t}^{f,(r)}, \sigma_f^2 \right)$
8:            Sample terminal node parameters: $\mathcal{M}_t^{f,(r)} \sim p\left( \mathcal{M}_t^f \mid \mathcal{T}_t^{f,(r)}, R_{1,-t}^{f,(r)}, \ldots, R_{n,-t}^{f,(r)}, \sigma_f^2 \right)$
9:            where $\sigma_f^2 = \begin{cases} 1, & \text{if } f = \mu \\ \sigma_M^2, & \text{if } f \in \{\mu_M, \tau_M\} \\ \sigma_Y^2, & \text{otherwise} \end{cases}$
10:       **end for**
11:    **end for**
12:    Sample residual variances:

$$(\sigma_M^2)^{(r)} \quad \sim \quad \text{Inv-Gamma}\left( a_{\sigma_M} + \frac{n}{2}, \ b_{\sigma_M} + \frac{1}{2} \sum_{i=1}^{n} (M_i - \hat{M}_i)^2 \right)$$

$$(\sigma_Y^2)^{(r)} \quad \sim \quad \text{Inv-Gamma}\left( a_{\sigma_Y} + \frac{n}{2}, \ b_{\sigma_Y} + \frac{1}{2} \sum_{i=1}^{n} (Y_i - \hat{Y}_i)^2 \right)$$

13:    Update $\boldsymbol{s}^{(r)}$ based on Eq. (8) and the M-H algorithm
14: **end for**

---

APPENDIX E: SIMULATION SETUP

In eight different scenarios, $P = 100$ potential confounders (except for Scenarios 5-7) are independently generated from $Unif(0,1)$ for $X_1 - X_2$ and from $N(0,1)$ for $X_3 - X_{100}$ where only 5 of them ($X_1 - X_5$) are true confounders:

- Scenario 1: Heterogeneous effects with non-linear terms.

$$
\begin{aligned}
P(A_i = 1) &= \Phi(0.5 + h_1(X_{i,1}) + h_2(X_{i,2}) - 0.5|X_{i,3} - 1| + 1.5X_{i,4}X_{i,5}) \\
M_i &\sim N(\mu_M(A_i, \boldsymbol{X}_i), 0.1^2) \\
\mu_M(A_i, \boldsymbol{X}_i) &= 1.3A_i - 0.5h_1(X_{i,1}) + 0.5h_2(X_{i,2}) + |X_{i,3} + 1| + 1.5X_{i,4} \\
&\quad - \exp(0.3X_{i,5}) - 1.5A_i|X_{i,5} + 0.3| \\
Y_i &\sim N(\mu_Y(A_i, M_i, \boldsymbol{X}_i), 0.3^2) \\
\mu_Y(A_i, M_i, \boldsymbol{X}_i) &= -A_i + h_1(X_{i,1}) + 1.5h_2(X_{i,2}) + 2|X_{i,3} + 1| - 1.5M_i + A_iX_{i,4} \\
&\quad + \exp(0.5X_{i,5})
\end{aligned}
$$

where $\Phi(\cdot)$ denotes the cumulative distribution function of the standard normal distribution.

- Scenario 2: Heterogeneous effects with non-linear terms with two additional predictors in $Y$ model.

$$
\begin{aligned}
P(A_i = 1) &= \Phi(0.5 + h_1(X_{i,1}) + h_2(X_{i,2}) - 0.5|X_{i,3} - 1| + 1.5X_{i,4}X_{i,5}) \\
M_i &\sim N(\mu_M(A_i, \boldsymbol{X}_i), 0.1^2) \\
\mu_M(A_i, \boldsymbol{X}_i) &= 1.3A_i - 0.5h_1(X_{i,1}) + 0.5h_2(X_{i,2}) + |X_{i,3} + 1| + 1.5X_{i,4} \\
&\quad - \exp(0.3X_{i,5}) - 1.5A_i|X_{i,5} + 0.3| \\
Y_i &\sim N(\mu_Y(A_i, M_i, \boldsymbol{X}_i), 0.3^2) \\
\mu_Y(A_i, M_i, \boldsymbol{X}_i) &= -A_i + h_1(X_{i,1}) + 1.5h_2(X_{i,2}) + 2|X_{i,3} + 1| - 1.5M_i + 2X_{i,4} \\
&\quad + \exp(0.5X_{i,5}) - 0.5A_i|X_{i,6}| - A_i|X_{i,7} + 1|
\end{aligned}
$$

- Scenario 3: Heterogeneous effects with non-linear terms with two additional instruments in $A$ model.

$$
\begin{aligned}
P(A_i = 1) &= \Phi(0.5 + h_1(X_{i,1}) + h_2(X_{i,2}) - 0.5|X_{i,3} - 1| + 1.5X_{i,4}X_{i,5} + 1.5|X_{i,6}| \\
&\quad - |X_{i,7} + 1|) \\
M_i &\sim N(\mu_M(A_i, \boldsymbol{X}_i), 0.1^2) \\
\mu_M(A_i, \boldsymbol{X}_i) &= 1.3A_i - 0.5h_1(X_{i,1}) + 0.5h_2(X_{i,2}) + |X_{i,3} + 1| + 1.5X_{i,4} - \exp(0.3X_{i,5}) \\
&\quad - 1.5A_i|X_{i,5} + 0.3| \\
Y_i &\sim N(\mu_Y(A_i, M_i, \boldsymbol{X}_i), 0.3^2) \\
\mu_Y(A_i, M_i, \boldsymbol{X}_i) &= -A_i + h_1(X_{i,1}) + 1.5h_2(X_{i,2}) + 2|X_{i,3} + 1| - 1.5M_i + 2X_{i,4} \\
&\quad + \exp(0.5X_{i,5})
\end{aligned}
$$

- Scenario 4: Heterogeneous effects with non-linear terms with two additional instruments in $A$ model and two additional instruments in $M$.

$$
\begin{aligned}
P(A_i = 1) &= \Phi(0.5 + h_1(X_{i,1}) + h_2(X_{i,2}) - 0.5|X_{i,3} - 1| + 1.5X_{i,4}X_{i,5} + 1.5|X_{i,6}| \\
&\quad - |X_{i,7} + 1|) \\
M_i &\sim N(\mu_M(A_i, \boldsymbol{X}_i), 0.1^2) \\
\mu_M(A_i, \boldsymbol{X}_i) &= 1.3A_i - 0.5h_1(X_{i,1}) + 0.5h_2(X_{i,2}) + |X_{i,3} + 1| + 1.5X_{i,4} - \exp(0.3X_{i,5}) \\
&\quad - 1.5A_i|X_{i,5} + 0.3| + 0.2|X_{i,8}| - 0.2|X_{i,9}| \\
Y_i &\sim N(\mu_Y(A_i, M_i, \boldsymbol{X}_i), 0.3^2) \\
\mu_Y(A_i, M_i, \boldsymbol{X}_i) &= -A_i + h_1(X_{i,1}) + 1.5h_2(X_{i,2}) + 2|X_{i,3} + 1| - 1.5M_i + 2X_{i,4} \\
&\quad + \exp(0.5X_{i,5})
\end{aligned}
$$

- Scenarios 5 & 6: Heterogeneous effects with non-linear terms in high-dimensional settings ($N < P = 500$ & $N < P = 2000$, respectively).

$$
\begin{aligned}
P(A_i = 1) &= \Phi(0.5 + h_1(X_{i,1}) + h_2(X_{i,2}) - 0.5|X_{i,3} - 1| + 1.5X_{i,4}X_{i,5} + 1.5|X_{i,6}| \\
&\quad - |X_{i,7} + 1|) \\
M_i &\sim N(\mu_M(A_i, \boldsymbol{X}_i), 0.1^2) \\
\mu_M(A_i, \boldsymbol{X}_i) &= 1.3A_i - 0.5h_1(X_{i,1}) + 0.5h_2(X_{i,2}) + |X_{i,3} + 1| + 1.5X_{i,4} - \exp(0.3X_{i,5}) \\
&\quad - 1.5A_i|X_{i,5} + 0.3| \\
Y_i &\sim N(\mu_Y(A_i, M_i, \boldsymbol{X}_i), 0.3^2) \\
\mu_Y(A_i, M_i, \boldsymbol{X}_i) &= -A_i + h_1(X_{i,1}) + 1.5h_2(X_{i,2}) + 2|X_{i,3} + 1| - 1.5M_i + 2X_{i,4} \\
&\quad + \exp(0.5X_{i,5})
\end{aligned}
$$

- Scenario 7: Heterogeneous effects with non-linear terms with $P = 20$.

$$
\begin{aligned}
P(A_i = 1) &= \Phi(0.5 + h_1(X_{i,1}) + h_2(X_{i,2}) - 0.5|X_{i,3} - 1| + 1.5X_{i,4}X_{i,5} + 1.5|X_{i,6}| \\
&\quad - |X_{i,7} + 1|) \\
M_i &\sim N(\mu_M(A_i, \boldsymbol{X}_i), 0.1^2) \\
\mu_M(A_i, \boldsymbol{X}_i) &= 1.3A_i - 0.5h_1(X_{i,1}) + 0.5h_2(X_{i,2}) + |X_{i,3} + 1| + 1.5X_{i,4} - \exp(0.3X_{i,5}) \\
&\quad - 1.5A_i|X_{i,5} + 0.3| \\
Y_i &\sim N(\mu_Y(A_i, M_i, \boldsymbol{X}_i), 0.3^2) \\
\mu_Y(A_i, M_i, \boldsymbol{X}_i) &= -A_i + h_1(X_{i,1}) + 1.5h_2(X_{i,2}) + 2|X_{i,3} + 1| - 1.5M_i + 2X_{i,4} \\
&\quad + \exp(0.5X_{i,5})
\end{aligned}
$$

In Scenarios, the first two variables are included in the models through non-linear functions $h_1(x) = (-1)^{I(x<0)}$ and $h_2(x) = (-1)^{I(x\geq 0)}$ and the remaining covariates are also added either with an absolute function or with an interaction term.

## APPENDIX F: ADDITIONAL SIMULATIONS

### F.1. SENSITIVITY ANALYSES

In the main manuscript, we conducted a sensitivity analysis of the prior specification by examining how performance varies with the number of trees used in the BART priors, and the results are presented in Table A1. In addition, we aim to further investigate how performance changes with

Table A1: Results (bias, absolute bias, mean squared error, and coverage) for sensitivity analysis with different numbers of trees.

| Numbers of trees in | BIAS | | Abs.BIAS | | MSE | | Coverage | |
|---|---|---|---|---|---|---|---|---|
| prognostic functions | DE | IE | DE | IE | DE | IE | DE | IE |
| 50 | 0.046 | -0.023 | 0.158 | 0.228 | 0.049 | 0.158 | 0.99 | 0.92 |
| 100 | 0.047 | -0.032 | 0.149 | 0.228 | 0.049 | 0.169 | 0.98 | 0.92 |

respect to different hyper-parameter values.

In our framework, we place a hyper-prior on $\alpha$ using the transformation $\alpha/(\alpha+P) \sim Beta(0.5, 1)$. This allows $\alpha$ to be adaptively tuned during the MCMC process. From our experiments, we observed that the value of $\alpha/P$, rather than $\alpha$ itself, plays a pivotal role in controlling the overall sparsity. Therefore, we believe that as long as either $\alpha$ becomes small or $P$ becomes large, the desired sparsity level can be achieved. This aligns well with the theoretical condition stated below Theorem 2: "The consistency is also guaranteed under the asymptotic growth condition $P = O(N^{-\gamma})$ with a properly controlled Dirichlet concentration parameter $\alpha$."

We further investigated how sensitive the model is to a fixed $\alpha$ by conducting an additional experiment based on the following scenario:

$$
\begin{aligned}
P(A_i = 1) &= \sigma(-0.3 + X_{i,1} - 1.5X_{i,2} + 0.8|X_{i,3} - 1| + 0.5X_{i,4}X_{i,5}) \\
M_i &\sim N(\mu_M(A_i, \boldsymbol{X}_i), 0.5^2) \\
\mu_M(A_i, \boldsymbol{X}_i) &= 1 + 0.9A_i + X_{i,1} - 0.5X_{i,2} + 0.5|X_{i,3} - 1| + 0.5X_{i,4}^2 - \sin(X_{i,5}) \\
Y_i &\sim N(\mu_Y(A_i, M_i, \boldsymbol{X}_i), 0.3^2) \\
\mu_Y(A_i, M_i, \boldsymbol{X}_i) &= X_{i,1} + 1.5X_{i,2} + 0.5X_{i,3}X_{i,4} - 0.4X_{i,5}^2 - 1.5M_i + A_i
\end{aligned}
$$

where $X_{i,j} \sim N(0,1)$ for $i = 1, \ldots, 250; j = 1, \ldots, P$ with $P = 100$. With the beta prior, the posterior mean of $\alpha$ was 1.49, and the posterior standard deviation was 0.22. Given $P = 100$, this implies $\alpha/P \approx 0.0149$. Now, we fixed $\alpha = 5$, which results in $\alpha/P = 0.05$, and examined the impact on the results. As shown in Table A2 of Appendix F.1, the overall results did not change substantially. This confirms that the absolute value of $\alpha$ is less critical than maintaining a sufficiently small ratio $\alpha/P \ll 1$, which effectively induces sparsity.

### F.2. UNMEASURED CONFOUNDER

Our assumption in Theorem 1 is likely to hold in high-dimensional settings with many observed covariates. However, we acknowledge that it may be more vulnerable in low-dimensional settings, where the chance of missing all relevant variables on a backdoor path increases. To further investigate, we empirically examined a violation of this assumption through an additional analysis based on Scenario 2, in which the true confounder $X_5$ is unmeasured.

As shown in Table A2 of F.2, when the assumption is violated due to an unmeasured confounder, the absolute bias of the estimated indirect effect (IE) was more than twice as large compared to the main Scenario 2. and the MSE increased by nearly a factor of four. We also observed a notable drop in coverage.

## F.3. MODEL MISSPECIFICATION

Given that our model is nonparametric, structural model misspecification is less of a concern. Instead, performance degradation is more likely when important variables are excluded (unmeasured). To assess this, we conducted an additional analysis based on Scenario 2, when a predictive but non-confounding variable $X_6$ is missing (i.e., outcome-model misspecification).

The results in Table A2 of F.3 show no noticeable difference from those of the main Scenario 2. That is, when an additional variable is a predictor but not a confounder, its exclusion from the model (i.e., even if model misspecification occurs) does not significantly impact overall performance. In contrast, when a true confounder is missing (as discussed in the previous subsection), we observe substantial performance degradation.

## F.4. COMPLEX BACKDOOR PATH

In this subsection, we clarify that our method employs a sparsity-inducing prior for the selection vector $s$, which naturally favors smaller confounder sets rather than maximal ones. If $X_i$ affects downstream variables $(A, M, \text{and/or } Y)$ only through $X_j$, our method tends to select either $X_j$ or $X_i$, typically the one with a stronger marginal effect. In other words, our method prefers to select direct causes of $A$, $M$, and/or $Y$, rather than general ancestors, when they are sufficient to block backdoor paths.

To illustrate this behavior empirically, we conducted a new experiment based on the following scenario:

$$
\begin{aligned}
P(A_i = 1) &= \sigma(-0.3 + X_{i,1} - 1.5X_{i,2} + 0.8|X_{i,3} - 1| + 0.5X_{i,4}X_{i,5}) \\
M_i &\sim N(\mu_M(A_i, \boldsymbol{X}_i), 0.5^2) \\
\mu_M(A_i, \boldsymbol{X}_i) &= 1 + 0.9A_i + X_{i,1} - 0.5X_{i,2} + 0.5|X_{i,9} - 1| + 0.5X_{i,10}^2 - \sin(X_{i,11}) \\
Y_i &\sim N(\mu_Y(A_i, M_i, \boldsymbol{X}_i), 0.3^2) \\
\mu_Y(A_i, M_i, \boldsymbol{X}_i) &= X_{i,1} + 1.5X_{i,2} + 0.5X_{i,9}X_{i,10} - 0.4X_{i,11}^2 - 1.5M_i + A_i
\end{aligned}
$$

where

- $X_6$ influences both $X_3$ and $X_9$,
- $X_7$ influences both $X_4$ and $X_{10}$,
- $X_8$ influences both $X_5$ and $X_{11}$.

The results are provided in Table A2 of F.4 and Figure A4. Our method successfully identified $X_1$, $X_2$, $X_9$, $X_{10}$ and $X_{11}$ as confounders, which are sufficient to block the backdoor paths related to $X_6 - X_8$. This provides strong empirical support that our method identifies a sparse set of direct causes as confounders.

Table A2: Results (bias, absolute bias, mean squared error, and coverage) for additional simulation scenarios (F.1-F.4)

| Scenario | Setting | BIAS | | Abs.BIAS | | MSE | | Coverage | |
|---|---|---|---|---|---|---|---|---|---|
| | | DE | IE | DE | IE | DE | IE | DE | IE |
| F.1 | $\alpha = 0.5$ | -0.023 | -0.042 | 0.096 | 0.173 | 0.021 | 0.056 | 0.99 | 0.99 |
| F.2 | unmeasured confounder | 0.052 | -0.038 | 0.426 | 1.080 | 0.316 | 1.878 | 0.82 | 0.60 |
| F.3 | model misspecification | -0.016 | 0.037 | 0.421 | 0.454 | 0.288 | 0.408 | 0.77 | 0.66 |
| F.4 | complex backdoor path | -0.016 | -0.003 | 0.059 | 0.130 | 0.008 | 0.032 | 0.99 | 0.99 |

## APPENDIX G: ACTG175 DATASET

The ACTG175 dataset, accessible through the R package `speff2trial`, comprises 27 variables. Of these, 15 pretreatment variables are identified as potential confounders, with descriptions based on the package documentation:

- `age` age in years at baseline

- `wtkg` weight in kg at baseline
- `hemo` hemophilia (0=no, 1=yes)
- `homo` homosexual activity (0=no, 1=yes)
- `drugs` history of intravenous drug use (0=no, 1=yes)
- `karnof` Karnofsky score (on a scale of 0-100)
- `oprior` non-zidovudine antiretroviral therapy prior to initiation of study treatment (0=no, 1=yes)
- `z30` zidovudine use in the 30 days prior to treatment initiation (0=no, 1=yes)
- `preanti` number of days of previously received antiretroviral therapy race race (0=white, 1=non-white)
- `gender` gender (0=female, 1=male)
- `str2` antiretroviral history (0=naive, 1=experienced)
- `strat` antiretroviral history stratification (1='antiretroviral naive', 2=' > 1 but 52 weeks of prior antiretroviral therapy', 3='> 52 weeks')
- `symptom` symptomatic indicator (0=asymptomatic, 1=symptomatic)
- `cd40` CD4Tcell count at baseline

APPENDIX H: EXTENSION TO MULTIPLE MEDIATORS

In practice, multiple mediators may exist. In such settings, researchers can isolate a mediator of primary scientific interest and interpret the indirect effect as that which is transmitted through this mediator. Effects transmitted through other paths are typically subsumed into the direct effect. This practice is also standard in the literature and provides meaningful interpretation even in complex settings.

That said, we also acknowledge the growing interest in mediation analysis involving multiple mediators. While our current theorems focus on the single-mediator case, the proposed model architecture is extensible. Specifically, building on the framework of Imai & Yamamoto (2013), we adapt our model to accommodate multiple causally independent mediators (with two mediators in this toy example) as follows

We construct separate mediator models (Eq. 6) for both mediators, $M_1$ and $M_2$

$$
\begin{aligned}
M_{1i} &= \mu_{M_1}(\boldsymbol{X}_i, \hat{\pi}_i) + \tau_{M_1}(\boldsymbol{X}_i)A_i + \epsilon_{1i} \\
M_{2i} &= \mu_{M_2}(\boldsymbol{X}_i, \hat{\pi}_i) + \tau_{M_2}(\boldsymbol{X}_i)A_i + \epsilon_{2i}
\end{aligned}
$$

The outcome model (Eq. 7) is expanded to include four clever covariates derived from the two mediator models. Both $M_1$ and $M_2$ are included as linear terms in the outcome regression model

$$
Y_i = \mu_Y(\boldsymbol{X}_i, \hat{\pi}_i, \hat{m}_{10i}, \hat{m}_{11i}, \hat{m}_{20i}, \hat{m}_{21i}) + \tau_Y(\boldsymbol{X}_i)A_i + \tau_Y'(\boldsymbol{X}_i)M_{1i} + \tau_Y''(\boldsymbol{X}_i)M_{2i} + \eta_i,
$$

where $\hat{m}_{jai} = \hat{\mathbb{E}}(M_{ji} \mid A_i = a, \boldsymbol{X}_i)$ for $j = 1, 2$. Separate selection vectors $\boldsymbol{s}^{M_1}$ and $\boldsymbol{s}^{M_2}$ are defined for the two mediators. The common selection vector $\boldsymbol{s}$ now includes parameters for all four clever covariates, resulting in a total dimension of $P + 5$

$$
\begin{aligned}
\boldsymbol{s}^A &= \left( \frac{s_1}{1 - (s_\pi + s_{m10} + s_{m11} + s_{m20} + s_{m21})}, \cdots, \frac{s_P}{1 - (s_\pi + s_{m10} + s_{m11} + s_{m20} + s_{m21})} \right) \\
\boldsymbol{s}^{M_1} &= \left( \frac{s_1}{1 - (s_{m10} + s_{m11} + s_{m20} + s_{m21})}, \cdots, \frac{s_P}{1 - (s_{m10} + s_{m11} + s_{m20} + s_{m21})}, \right. \\
&\qquad \left. \frac{s_\pi}{1 - (s_{m10} + s_{m11} + s_{m20} + s_{m21})} \right) \\
\boldsymbol{s}^{M_2} &= \left( \frac{s_1}{1 - (s_{m10} + s_{m11} + s_{m20} + s_{m21})}, \cdots, \frac{s_P}{1 - (s_{m10} + s_{m11} + s_{m20} + s_{m21})}, \right. \\
&\qquad \left. \frac{s_\pi}{1 - (s_{m10} + s_{m11} + s_{m20} + s_{m21})} \right) \\
\boldsymbol{s}^Y &= (s_1, \cdots, s_P, s_\pi, s_{m10}, s_{m11}, s_{m20}, s_{m21}).
\end{aligned}
$$

To empirically evaluate this extension, we implemented the model in a synthetic setting involving two causally independent mediators (with 100 potential confounders $\boldsymbol{X}'_i s$). Specifically, the data-generating models for the two mediators and the outcome are as follows

$$
\begin{aligned}
P(A_i = 1) &= \sigma(-0.3 + X_{i,1} - 1.5X_{i,2} + 0.8|X_{i,3} - 1| + 0.5X_{i,4}X_{i,5}) \\
M_{1i} &\sim N(\mu_{M_1}(A_i, \boldsymbol{X}_i), 0.5^2) \\
\mu_{M_1}(A_i, \boldsymbol{X}_i) &= 1 + 1.2A_i + X_{i,1} - 0.5X_{i,2} + 0.5|X_{i,3} - 1| - 0.5X_{i,4}^2 - \sin(X_{i,5}) \\
M_{2i} &\sim N(\mu_{M_2}(A_i, \boldsymbol{X}_i), 0.5^2) \\
\mu_{M_2}(A_i, \boldsymbol{X}_i) &= 1 + 1.5A_i + 0.5X_{i,1} - X_{i,2} + 0.75|X_{i,3} - 1| - 0.4X_{i,4}^2 - \sin(X_{i,5}) \\
Y_i &\sim N(\mu_Y(A_i, M_i, \boldsymbol{X}_i), 0.3^2) \\
\mu_Y(A_i, M_i, \boldsymbol{X}_i) &= X_{i,1} + 1.5X_{i,2} + 0.5X_{i,3}X_{i,4} - 0.4X_{i,5}^2 - M_{1i} - 0.7M_{2i} + A_i.
\end{aligned}
$$

The true indirect effects were set to $-2.25$ and $1.0$, respectively. Table A3 summarizes the result. These results confirm that the proposed framework can be naturally extended to multiple-mediator

Table A3: Results (bias, absolute bias, mean squared error, and coverage) for the scenario with multiple mediators (two mediators).

|  | BIAS | | Abs.BIAS | | MSE | | Coverage | |
|---|---|---|---|---|---|---|---|---|
|  | DE | IE | DE | IE | DE | IE | DE | IE |
| Multiple | 0.302 | -0.354 | 0.315 | 0.380 | 0.126 | 0.202 | 0.58 | 0.97 |

settings without requiring fundamental change to the algorithm structure. As the number of mediators increases, variable importance from individual mediator models may dominate the overall selection probability vector. To address this, we plan to extend the framework to a more general setting with multiple mediators in future work.

APPENDIX H: TABLES AND FIGURES

Table A4: Operating characteristics (true positive rate, false positive rate, positive predictive value, and F1 score) for different scenarios: Scenario 1 (base), Scenario 2 (with two predictors), Scenarios 3 and 4 (with two instruments in exposure and mediator models, respectively), Scenario 5 ($N < P = 500$), and Scenario 7 ($P = 20$). Selection results are reported separately for the treatment model ($A$), mediator model ($M$), and outcome model ($Y$).

| Scenario | Method | TPR | | | FPR | | | PPV | | | F1 | | |
|---|---|---|---|---|---|---|---|---|---|---|---|---|---|
| | | $A$ | $M$ | $Y$ | $A$ | $M$ | $Y$ | $A$ | $M$ | $Y$ | $A$ | $M$ | $Y$ |
| | BCMF-CS | 1 | 1 | 1 | 0.00 | 0.00 | 0.00 | 1 | 1 | 1 | 1 | 1 | 1 |
| (1) | BCMF | 1 | 1 | 1 | 1 | 0.72 | 0.98 | 0.05 | 0.07 | 0.05 | 0.09 | 0.13 | 0.09 |
| | DCMF | 0.44 | 1 | 0.82 | 0.00 | 0.00 | 0.00 | 0.82 | 1 | 0.99 | 0.55 | 1 | 0.89 |
| | BCMF-CS | 1 | 1 | 1 | 0.01 | 0.01 | 0.01 | 0.83 | 0.83 | 0.83 | 0.91 | 0.91 | 0.90 |
| (2) | BCMF | 1 | 1 | 1 | 1 | 0.72 | 0.98 | 0.05 | 0.07 | 0.05 | 0.09 | 0.13 | 0.09 |
| | DCMF | 0.45 | 1 | 0.87 | 0.01 | 0.00 | 0.01 | 0.85 | 1 | 0.77 | 0.57 | 1 | 0.81 |
| | BCMF-CS | 1 | 1 | 1 | 0.00 | 0.00 | 0.00 | 1 | 1 | 1 | 1 | 1 | 1 |
| (3) | BCMF | 1 | 1 | 1 | 1 | 0.72 | 0.98 | 0.05 | 0.07 | 0.05 | 0.09 | 0.13 | 0.09 |
| | DCMF | 0.41 | 1 | 0.87 | 0.02 | 0.00 | 0.00 | 0.51 | 1 | 0.99 | 0.45 | 1 | 0.93 |
| | BCMF-CS | 1 | 1 | 1 | 0.00 | 0.00 | 0.00 | 0.95 | 0.94 | 0.95 | 0.97 | 0.96 | 0.97 |
| (4) | BCMF | 1 | 1 | 1 | 1 | 0.82 | 0.97 | 0.05 | 0.06 | 0.05 | 0.10 | 0.11 | 0.10 |
| | DCMF | 0.41 | 1 | 0.99 | 0.03 | 0.01 | 0.00 | 0.51 | 0.83 | 0.99 | 0.44 | 0.91 | 0.99 |
| | BCMF-CS | 1 | 1 | 1 | 0.00 | 0.00 | 0.00 | 0.99 | 1 | 0.99 | 0.99 | 1 | 0.99 |
| (5) | BCMF | 0.28 | 1 | 0.99 | 0.01 | 0.01 | 0.02 | 0.76 | 0.80 | 0.73 | 0.43 | 0.88 | 0.83 |
| | DCMF | 0.09 | 1 | 0.92 | 0.00 | 0.00 | 0.00 | 0.92 | 1 | 0.99 | 0.35 | 1 | 0.95 |
| | BCMF-CS | 1 | 1 | 1 | 0.00 | 0.00 | 0.00 | 0.99 | 0.99 | 0.99 | 0.99 | 1 | 0.99 |
| (7) | BCMF | 0.29 | 1 | 0.99 | 0.01 | 0.02 | 0.02 | 0.79 | 0.80 | 0.71 | 0.45 | 0.88 | 0.82 |
| | DCMF | 0.11 | 1 | 0.93 | 0.00 | 0.00 | 0.00 | 0.91 | 1 | 0.99 | 0.34 | 1 | 0.96 |

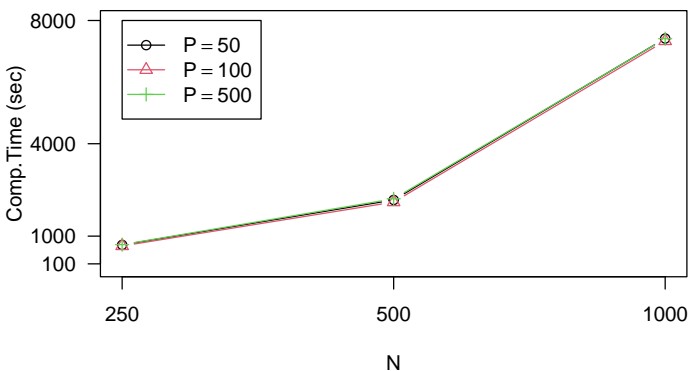

Figure A1: Computational time as a function of the number of observations ($N$) and potential confounders ($P$).

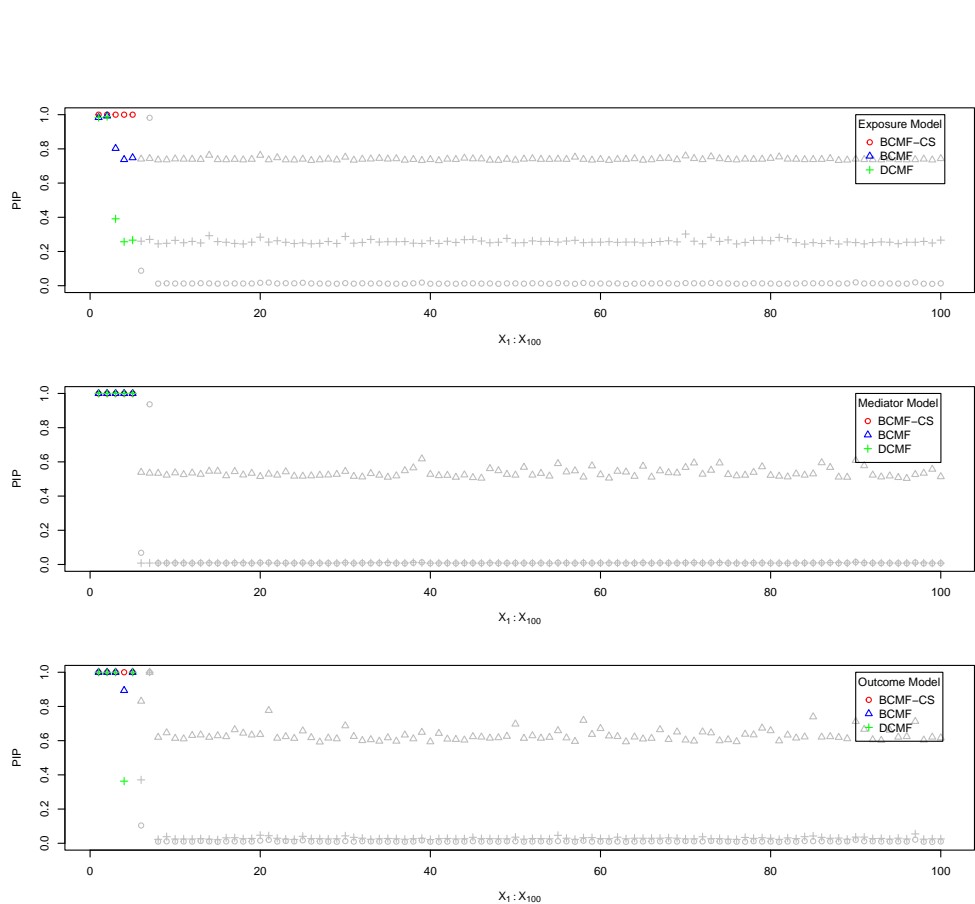

Figure A2: [Scenario 2] Posterior inclusion probabilities (PIP) from three competing methods (BCMF-CS, BCMF, and DCMF) for the exposure, mediator, and outcome models. The true confounders are represented by colored points, while the noise variables are depicted as grey points.

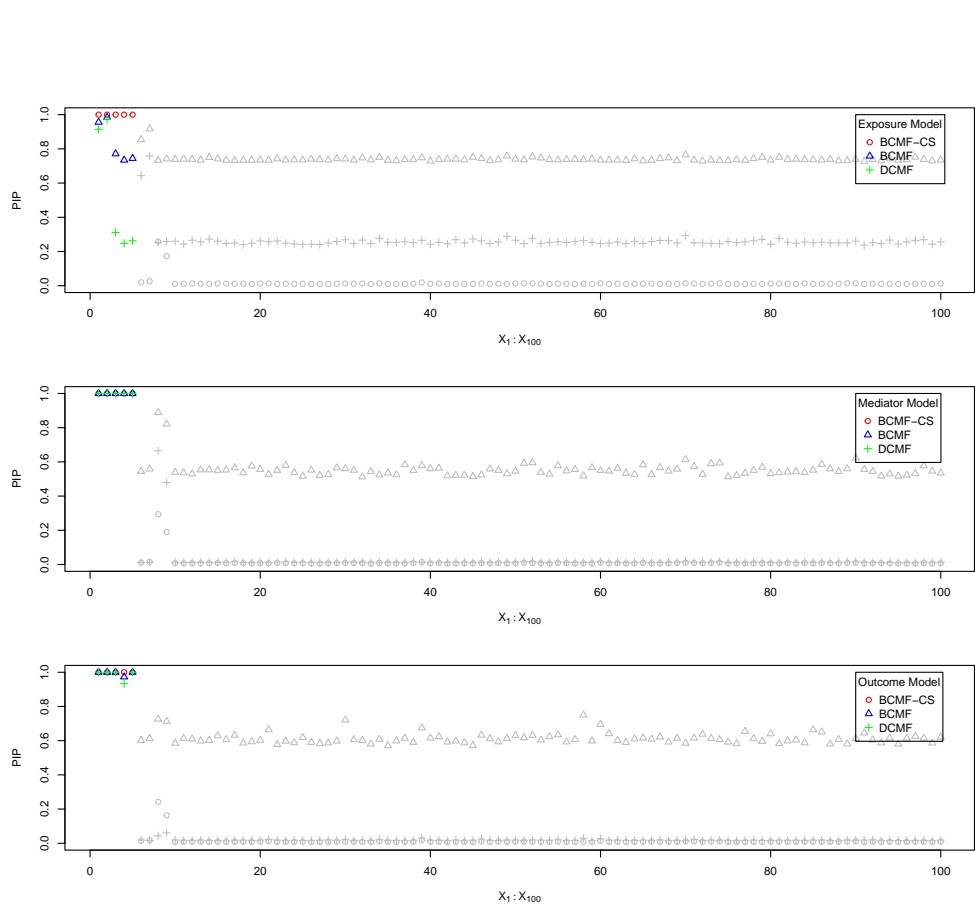

Figure A3: [Scenario 4] Posterior inclusion probabilities (PIP) from three competing methods (BCMF-CS, BCMF, and DCMF) for the exposure, mediator, and outcome models. The true confounders are represented by colored points, while the noise variables are depicted as grey points.

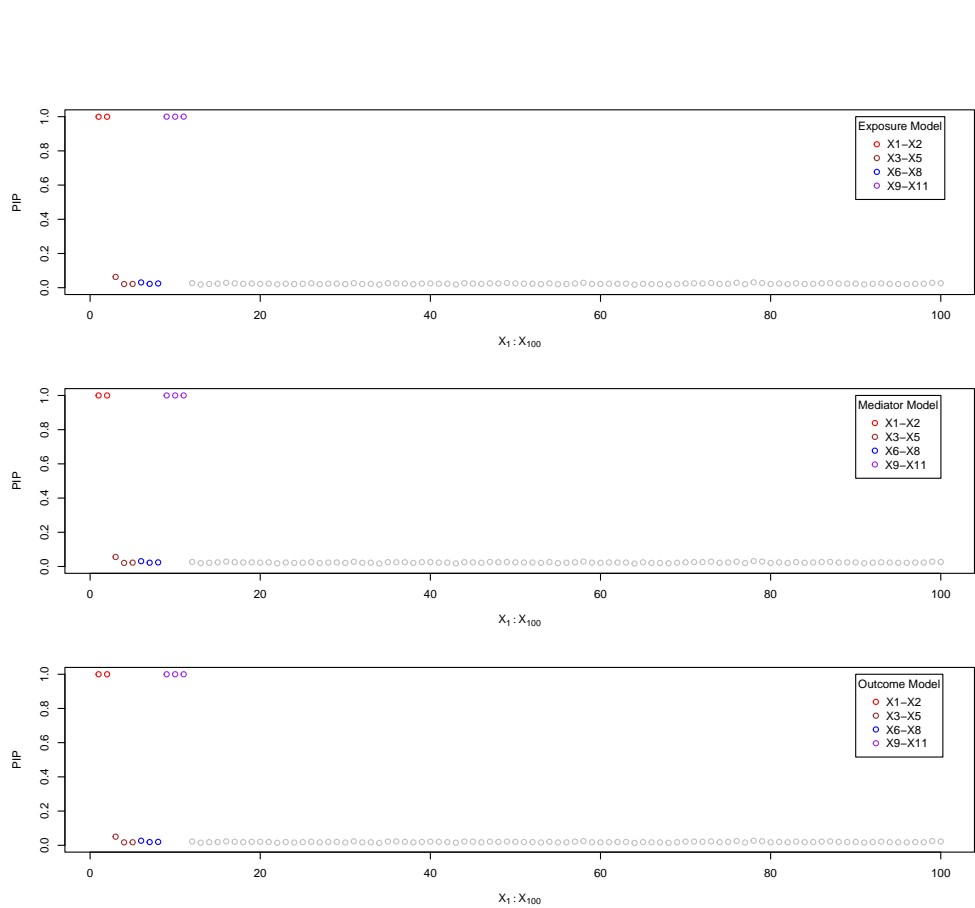

Figure A4: [Scenario F.4] Posterior inclusion probabilities (PIP) for the exposure, mediator, and outcome models under the complex backdoor path structure. The true confounders are represented by colored points, while the noise variables are depicted as grey points.

