# OpenReview forum: "Bayesian Decision Trees for Confounder Selection in Mediation Analysis"
_ICLR.cc/2026/Conference — ICLR 2026 Conference Withdrawn Submission_

### Official Review · Reviewer_HvKq · 2025-10-26

**Soundness:** 3
**Presentation:** 4
**Contribution:** 3
**Rating:** 4
**Confidence:** 3

**Summary:**

The paper studies confounder selection for causal mediation analysis with high-dimensional covariates. It introduces a “mediation disjunctive cause” criterion that extends standard disjunctive cause ideas to the exposure–mediator–outcome setting and shows that sets satisfying this criterion support sequential ignorability, given rich pretreatment covariates. Methodologically, the paper proposes a Bayesian causal mediation forest built from BART components for the exposure, mediator, and outcome, linked by a common sparsity prior on variable selection probabilities to favor variables that matter across all three models. The theory gives selection consistency (including regimes with $P>N$ under growth conditions) and posterior consistency of direct and indirect effects when the models are well specified. Experiments across diverse scenarios and one clinical dataset (ACTG175) suggest improved effect estimation and more targeted selection relative to baselines.

**Strengths:**

The work's main idea is original in scope: it formulates a mediation-specific selection principle that acknowledges three types of confounding ($\mathrm{A}-\mathrm{M}, \mathrm{A}-\mathrm{Y}, \mathrm{M}-\mathrm{Y}$) and ties it to identification. That fills a gap left by criteria developed for treatment-outcome problems without mediators. The modeling choice-three linked BART components with a shared Dirichlet prior on selection probabilities-offers a clear mechanism to surface variables that play confounding roles in more than one submodel, which is exactly what mediation requires. On technical quality, the paper states conditions under which selection is consistent (including $P$ growing with $N$) and links correct selection to consistent estimation of direct and indirect effects; the assumptions are explicit and the statements align with the modeling choices. Empirically, the simulation design stresses instruments, nuisance predictors, and high-dimensional regimes; the method shows lower bias/MSE and better coverage in many settings, which supports the practical value of joint selection. Clarity is generally good: notation is standard, the three submodels are written down, and the algorithmic idea (reparameterizing the selection vectors through a single Dirichlet prior) is explained. The potential significance is high because mediation analyses in practice often face large covariate sets and need principled, data-driven adjustment that guards against instruments and unrelated predictors while remaining computationally feasible.

**Weaknesses:**

1. **Assumptions**. Selection consistency and effect consistency rely on correct specification of the exposure, mediator, and outcome models and on regularity conditions (including Dirichlet concentration behavior). In applied settings, BART components can still be misspecified (e.g., heteroscedastic mediator or non-Gaussian noise). So I am willing to see some robustness results or at least stress tests where one or more components are misspecified (e.g., non-additive interactions only present in $M$ or $Y$). In particular, how selection and effect errors change when the Dirichlet concentration parameter is mis-tuned?

2. **Finite-sample guidance on the prior**. The reparameterized prior is elegant, but the paper offers limited guidance for setting $\alpha$ and tree counts across submodels beyond defaults. Empirically, the authors conducted a sensitivity analysis of the prior specification in Section F.1. But I am curious if there are any theoretical results which map sample size and $P$ to recommended $\alpha$, number of trees, and depth priors?

3. **Evaluation design**. The simulation design is broad, but two important regimes are under-explored. First, mediator–outcome confounders partly affected by exposure (post-treatment common causes) violate sequential ignorability and are common in practice. Can the authors show how the method behaves and whether it tends to select such variables (and what that does to bias)? Second, highly correlated covariates and ancestor/proxy chains can cause a high variable inclusion between parents and proxies. Can the authors include correlation-structured designs and report stability (e.g., PIP dispersion across correlated groups)?

4. **Real-data study**. The ACTG175 application uses an RCT. Because randomization weakens A–M and A–Y confounding by design, it is not the strongest test of a confounder-selection method for observational mediation.

**Questions:**

1. Misspecification and sensitivity. How does performance change if the mediator model has heavy tails or heteroscedastic errors, or if the exposure model is probit vs. logit while fitted with a probit link? Please add stress tests and report selection and effect errors.

2. Correlation and proxies. When two or more covariates are highly correlated and sit on the same backdoor path, does the shared Dirichlet prior pick one stably, or does inclusion probability split across them?

3. Tuning the Dirichlet concentration. Provide a data-driven rule or cross-validation scheme for $\alpha$. Right now, users have little guidance beyond defaults. Could AIC/LOO select $\alpha$ by targeting coverage or MSE of effects?

4. Sequential ignorability violations. Please include a simulation where some mediator–outcome confounders are affected by treatment. Even a small violation can create bias; showing failure modes and diagnostic signals would help practitioners.

5. MCMC diagnostics. Please provide effective sample sizes, acceptance rates for grow/prune/change steps, and trace plots for key quantities (e.g., average PIP, average effects). This will build confidence in the numerical results.

6. Interpretability. The authors claimed that the shared prior encourages “direct causes” over upstream ancestors (Line 246). Can you formalize this tendency? A small proposition or lemma with assumptions would clarify when the method prefers nearer causes along backdoor paths.

7. On low coverage in Scenario 2 (heterogeneous effects; non-linear outcome with two extra outcome predictors).
In Scenario 2 the reported coverage is low for all methods, with your approach also under-covering (e.g., BCMF-CS 0.64; BCMF 0.17; DCMF 0.22; LSEM 0.38). Could you explain the mechanism behind this failure and provide evidence that separates bias from variance underestimation? Please indicate which assumption is most likely at fault and what diagnostic a practitioner should use to detect it.

---

### Official Review · Reviewer_k7PE · 2025-10-28

**Soundness:** 3
**Presentation:** 2
**Contribution:** 2
**Rating:** 4
**Confidence:** 4

**Summary:**

This paper proposes a Bayesian decision tree framework for confounder selection in mediation analysis, addressing the challenge of identifying variables that confound the exposure–mediator–outcome relationships in high-dimensional observational data. The method, called BCMF-CS, is designed to identify relevant confounders from a high-dimensional pool of covariates. The paper establishes theoretical properties for the method and demonstrates its superior performance in simulation studies and a real-world application.

**Strengths:**

This study introduces the Mediation Disjunctive Cause Criterion, addressing a critical gap in causal mediation analysis.

It further presents a data-driven approach for confounder selection specifically designed for mediation settings.

Comprehensive simulations under heterogeneous, nonlinear, and high-dimensional conditions highlight the superior performance of the proposed BCMF-CS model.

**Weaknesses:**

The theoretical guarantees rely heavily on the provided mediator and the pretreatment variable assumptions, which are strong and often unverifiable in practice.

The paper’s claim to be the first to propose a data-driven method for confounder selection in mediation analysis seems somewhat overstated. Several related works have already explored data-driven or algorithmic approaches to identifying valid adjustment sets and confounders. For instance, the authors may wish to consider the following studies:

>Complete Graphical Characterization and Construction of Adjustment Sets in Markov Equivalence Classes of Ancestral Graphs

>Separators and Adjustment Sets in Causal Graphs: Complete Criteria and an Algorithmic Framework

>Causal mediation analysis with double machine learning

>Toward Unique and Unbiased Causal Effect Estimation from Data with Hidden Variables

These works provide important perspectives on graphical and representation-based methods for confounder identification, and acknowledging them would help better position the paper’s contribution.


Figure 1 presents an interesting example. The three variables $X_1$, $X_2$, and $X_3$ seem distinguishable through standard conditional independence tests. Why, then, is BART necessary in this context? What specific advantages does the proposed BART-based approach offer over simpler conditional independence–based methods?

The pretreatment assumption is not clearly stated. As shown in Figure 1, $X_2$ does not appear to satisfy this assumption, since it seems to be a pretreatment variable for $M$ rather than for $A$. It would be helpful if the authors could clarify and formalize this assumption more precisely.

The proposed method is not evaluated against several recent and relevant approaches, such as Disentangled Representation for Causal Mediation Analysis and Causal Mediation Analysis with Hidden Confounders. In the absence of these comparisons, it is difficult to convincingly demonstrate the superiority or practical advantages of the proposed algorithm over the current state of the art.

**Questions:**

Since BART is a nonparametric ensemble method, how do the authors ensure interpretability or stability of the selected confounder set across MCMC runs?

---

### Official Review · Reviewer_HnKB · 2025-11-01

**Soundness:** 3
**Presentation:** 4
**Contribution:** 3
**Rating:** 6
**Confidence:** 3

**Summary:**

This paper addresses a critical gap in causal mediation analysis: the lack of data-driven methods for confounder selection in the exposure-mediator-outcome framework. This paper introduces the Mediation Disjunctive Cause Criterion, which generalizes traditional criteria of confounder selection to mediation contexts. A data-driven method built on Bayesian Additive Regression Trees (BART) is proposed, which is theoretically and empirically superior than baselines.

**Strengths:**

1. This paper’s originality is multifaceted, spanning theoretical framing, methodological innovation, and practical adaptation—moving beyond incremental tweaks to resolve longstanding gaps in the field.
2. This paper exhibits exceptional quality, with theoretical rigor, robust experimental design, and transparency
3. This paper excels in clarity, balancing technical precision with readability—a rare feat for work at the intersection of causal inference and Bayesian ML.

**Weaknesses:**

1. This paper acknowledges that unmeasured confounding degrades performance (Appendix F.2) but provides no solutions to mitigate this.
2. The core theoretical results and main experiments focus exclusively on single-mediator models, despite that "multiple or sequential mediators frequently arise in practice" is acknowledged.
3. The paper claims computational efficiency but only validates runtime for limited number of samples and covariates.

**Questions:**

Would the criterion still work with a non-BART model such as random forests?

---

### Official Review · Reviewer_Wh2n · 2025-11-05

**Soundness:** 3
**Presentation:** 2
**Contribution:** 1
**Rating:** 2
**Confidence:** 4

**Summary:**

This paper provides a BART-based confounder selection method in mediation setting in causal inference. It extends existing BART-based variable selection methods (e.g., Kim et al. 2023) to the exposure–mediator–outcome causal structure, where confounding can appear in three relationships (A–M, A–Y, M–Y).

**Strengths:**

The paper is theoretically well-grounded. The theoretical development is rigorous and clearly connects identification assumptions with the proposed Bayesian framework.

**Weaknesses:**

This work overlooks the established causal graph–based foundation of mediation analysis. Mediation analysis, as developed by Pearl, Robins, Greenland, and others, is fundamentally graphical, because its assumptions are meant to be explicitly encoded through causal diagrams, enabling transparent identification of valid adjustment sets.

A rich line of research (e.g., Tian et al., 1998, Shpitser et al., 2010, 2012, Entner et al., 2013, Perković et al., 2018, Wang et al., 2023) already provides formal algorithms for identifying such confounder sets from a known graph, or from its Markov equivalence class when the graph is only partially known. In contrast, this paper treats confounder selection as a purely statistical variable selection task, detached from the causal structure that defines the problem.

Even when the graph is unknown, existing methods (e.g., Entner et al., 2013; Perković et al., 2018) exploit conditional independences to recover valid adjustment sets, which naturally extend to mediation settings (A–M, M–Y, A–Y). As such, it is unclear where this work fits conceptually, since mediation analysis is inherently graph-based and already admits principled solutions for confounder identification both with and without explicit graph knowledge.

Overall, I do not see this problem as theoretically or practically valid. Mediation analysis is graph-foundational. With a graph, we already have established solutions for identifying confounders; and even without a graph, conditional independence structure provides principled ways to determine valid adjustment sets.

**Questions:**

Please see Weakness.

---

### Note · Authors · 2025-11-16

I have read and agree with the venue's withdrawal policy on behalf of myself and my co-authors.